# Horizontal acquisition of a DNA ligase improves DNA damage tolerance in eukaryotes

Emilien Nicolas [1] ✉, Paul Simion [2,3], Marc Guérineau[1], Matthieu Terwagne [2], Mathilde Colinet[2], Julie Virgo[2], Maxime Lingurski[1], Anaïs Boutsen[2], Marc Dieu [4], Bernard Hallet [5] ✉ & Karine Van Doninck [1,2] ✉

Bdelloid rotifers are part of the restricted circle of multicellular animals that can withstand a wide range of genotoxic stresses at any stage of their life cycle. In this study, bdelloid rotifer *Adineta vaga* is used as a model to decipher the molecular basis of their extreme tolerance. Proteomic analysis shows that a specific DNA ligase, different from those usually involved in DNA repair in eukaryotes, is strongly over-represented upon ionizing radiation. A phylogenetic analysis reveals its orthology to prokaryotic DNA ligase E, and its horizontal acquisition by bdelloid rotifers and plausibly other eukaryotes. The fungus *Mortierella verticillata*, having a single copy of this DNA Ligase E homolog, also exhibits an increased radiation tolerance with an over-expression of this DNA ligase E following X-ray exposure. We also provide evidence that *A. vaga* ligase E is a major contributor of DNA breaks ligation activity, which is a common step of all important DNA repair pathways. Consistently, its heterologous expression in human cell lines significantly improves their radio-tolerance. Overall, this study highlights the potential of horizontal gene transfers in eukaryotes, and their contribution to the adaptation to extreme conditions.

All living organisms endure constant endogenous and exogenous stresses. Under a stressful environment, the survival potential of an individual depends on its capacity of protecting its cellular components from those threats and on its ability to repair the incurred damages. Preservation of genome integrity is a fundamental process for cell homeostasis that depends on a global signaling network referred to as DNA damage response (DDR) (e.g., see refs. 1,2). DDR detects different types of DNA damages and coordinates a global response that includes cell cycle control, activation of transcription, induction of DNA repair pathways, senescence, and cell death[1,2]. The outcome of the DDR depends on several factors including the amount and the complexity of DNA damages[1,3].

Under similar DNA damaging conditions, survival capacity varies greatly among living organisms, some being highly sensitive and others being highly tolerant. This continuum reflects the evolution of mechanisms involved in DNA protection and repair, which are more efficient and sometimes unique in certain organisms compared to others (e.g., see refs. 3,4). Organisms that are tolerant to extremely harsh environments are often found among bacteria and archaea, with the well-known example of *Deinococcus radiodurans*[5], which tolerates complete desiccation and over 16 kilograys (kGy) of ionizing radiation (IR). Desiccation and IR both generate a wide range of DNA damage, from base lesions to single- and double-strand breaks (SSBs and DSBs) that generally cause cell death in animals. A few remarkable animals however survive complete desiccation and extreme doses of IR,

[1]Université Libre de Bruxelles, Molecular Biology and Evolution, Brussels 1050, Belgium. [2]Université de Namur, Laboratory of Evolutionary Genetics and Ecology, Namur 5000, Belgium. [3]Université de Rennes, Ecosystèmes, biodiversité, évolution (ECOBIO UMR 6553), CNRS, Rennes, France. [4]Université de Namur, MaSUN-mass spectrometry facility, Namur 5000, Belgium. [5]Université Catholique de Louvain, Louvain Institute of Biomolecular Science and Technology, Louvain-la-Neuve 1348, Belgium. ✉e-mail: emilien.nicolas@gmail.com; bernard.hallet@uclouvain.be; karine.van.doninck@ulb.be

exhibiting a high DNA damage tolerance[6]. The most notorious of those are tardigrades and bdelloid rotifers[7–18].

Genomic and proteomic studies performed on the tardigrade *Ramazzottius varieornatus*, identified a unique DNA protecting protein termed Dsup[19–21]. This highly charged protein binds to chromatin and decreases the DNA fragmentation induced by X-ray radiation or by reactive oxygen species (ROS)[19–21]. As a consequence, human cells expressing Dsup acquired higher tolerance to radiation[19]. This discovery pinpoints that stress tolerance may involve specialized and species-specific proteins, emphasizing the potential for identifying novel players involved in DNA damage tolerance or repair from stress resistant organisms.

Bdelloid rotifers are microscopic animals living in semi-terrestrial habitats where they experience frequent desiccation at any stage of their life cycle[13,22]. This resistance to desiccation is often correlated to a strong tolerance to ionizing radiation with certain species like *Adineta vaga* being able to withstand doses up to 3 kGy of X-ray, proton or Fe radiation[10,16,23,24]. At such high radiation doses bdelloid rotifers experience considerable DNA damage including genome fragmentation due to DSBs, while still surviving, exhibiting a high DNA damage tolerance[10,13]. Following radiation, the organisms manage to recover DNA integrity by a particularly efficient DNA repair process that reassembles fragmented chromosomes[13,16]. Maintaining such an active DNA repair process under a highly stressful environment correlates with a better protection of proteins, as shown by a lower level of protein carbonylation detected in irradiated *A. vaga* compared to *Caenorhabditis elegans*[11].

We recently produced the first chromosome-scale genome assembly of *A. vaga* providing unprecedented avenues to identify key stress tolerance genes of bdelloid rotifers and developed the tool Alienomics, confirming the high number of horizontally acquired genes in these organisms[25]. Indeed, previous studies have demonstrated that bdelloid rotifers harbor a large amount of foreign genes coming from bacteria, fungi and plants, often clustered in the sub-telomeric regions along with transposable elements and reaching >6% across studied bdelloid genomes[26–29]. In order to pinpoint the key actors of their extreme radiation tolerance, a proteomic analysis was conducted here on *A. vaga* following exposure to ionizing radiation. This analysis revealed that irradiation mainly activates the production of proteins involved in DNA repair, including a specific and highly over-represented DNA ligase, previously identified as LigK[30]. We provided evidence that this enzyme, renamed AvLigE, has strong homologies with the ligase E family of prokaryotic DNA ligases, and that it was horizontally acquired by bdelloid rotifers to act as a leading actor in the DDR pathway. We propose that a similar scenario could have occurred in different eukaryotic lineages, including the fungus species *Mortierella verticillata*, in which induction of MvLigE ligase was found to correlate with increased radio-tolerance. Our results therefore highlight the importance of horizontal gene transfers between prokaryotes and eukaryotes for the adaptation to harsh conditions. Consistently, the heterologous expression of AvLigE in human cells was shown here to significantly improve radio-tolerance.

## Results

### Over-representation of a specific DNA ligase upon irradiation of *A. vaga*

Comparative proteomic analysis was performed upon irradiation of hydrated *A. vaga* individuals to 1 kGy of X-rays. At such a high IR dose, the genome of *A. vaga* is fragmented into small pieces (between 200 and 750 kbp) that are progressively re-assembled into larger fragments, indicative of an active DNA repair process being triggered by DNA damages within 24 h post-irradiation (Supplementary Fig. S1A). Whole protein extracts were prepared from non-irradiated and irradiated *A. vaga* individuals at different time points of recovery (i.e., t0h,

t4h, t24h, and t72h), and the proteins were identified and quantified by mass spectrometry.

No proteins were shown to be up- or down-regulated immediately after irradiation (t0h), indicating that a certain delay is required to adapt to the stress and start DNA repair. However, after 4h of recovery (t4h), several proteins mainly involved in DNA repair were significantly over-represented, including a DNA ligase, the DNA polymerase β, a polynucleotide kinase/phosphatase (PNKP), and PARP2 (Fig. 1a and Supplementary Table S1). These proteins are known to be specifically involved in the base excision repair (BER) pathway, consistent with base lesions being a major DNA damage occurring upon X-ray radiation[31,32]. The early expression of DNA actors following IR correlates with PFGE data showing that re-assembly of high molecular weight DNA fragments seems to mainly occur within the first 24 h of recovery (Supplementary Fig. S1A)[13]. Another protein, the alpha-protein kinase vwka, was also over-represented in the irradiated samples. This protein contributes to myosin-mediated cellular contraction processes in protozoa[33], and is thus likely required for muscle contraction in *A. vaga*, a function that is visually affected in irradiated bdelloid rotifers, which present a clear spasmatic-like contraction phenotype. All these proteins remained over-represented at t24h and t72h post-irradiation, consistent with sustained DNA repair activity (Fig. 1a). However, other protein families were also over-represented upon prolonged recovery (t72h), which likely reflects a more general adaptation of irradiated cells in the long term (Fig. 1a). Similar results were independently obtained after irradiation of hydrated *A. vaga* at 0.8 kGy of X-rays, showing that the determinants of DNA repair response remain consistent (Supplementary Fig. S1B).

Among the over-represented proteins, the DNA ligase was by far the one with the highest induction level (6- to 7-fold). This ligase was also shown by transcriptomic analysis[30] to be strongly induced upon desiccation treatment and was characterized into further details here. The bdelloid rotifer *A. vaga* has a diploid genome structure with signatures of paleo-tetraploidy, and as 40% of the genes, this DNA ligase gene is present in four copies[25]. Two copies are located on homologous chromosomes 4a and 4b (copies A) and two other copies on homologous chromosomes 1a and 1b (copies B). Homoeologous ligase copies A and B of *A. vaga* are divergent in sequence (68.5% of identity at the amino acid level) and only peptides from the copies B were detected in *A. vaga* by mass spectrometry, suggesting a divergence in expression between homoeologous copies.

Expression of both proteins was also examined by Western-blotting using specific antibodies of each variant (Supplementary Fig. S1C). Consistent with the mass spectrometry results, copy B was barely detected in non-irradiated samples, but strongly accumulated over time after irradiation (Fig. 1b). Conversely, no such accumulation was observed for copy A ligase which remained at a low level of expression both before and after IR treatment (t24h, Fig. 1b). This result thus confirms that only the copy B of the DNA ligase is induced in response to irradiation stress in *A. vaga*, suggesting a specific role in DNA repair.

Immuno-fluorescence detection of DNA ligase B in *A. vaga* individuals revealed an increased nuclear signal starting from t24h post-irradiation (Fig. 1c and Supplementary Fig. S1D). However, virtually no DNA ligase B labeling was observed in the middle part of the adult body where the ovaries and germline are localized (circled by dashed lines in Fig. 1c and Supplementary Fig. S1D). The IR-induced expression pattern of this DNA ligase B and its nuclear localization in *A. vaga*, strongly suggests that it might be a key actor involved in bdelloid rotifer DNA repair in somatic nuclei.

### Evidence for a bacterial origin and horizontal transfer of the IR-induced DNA ligase

Bdelloid rotifers represent the metazoan clade with the highest proportion of foreign genes in their genome (up to 8% in *A. vaga*[25,28,29]).

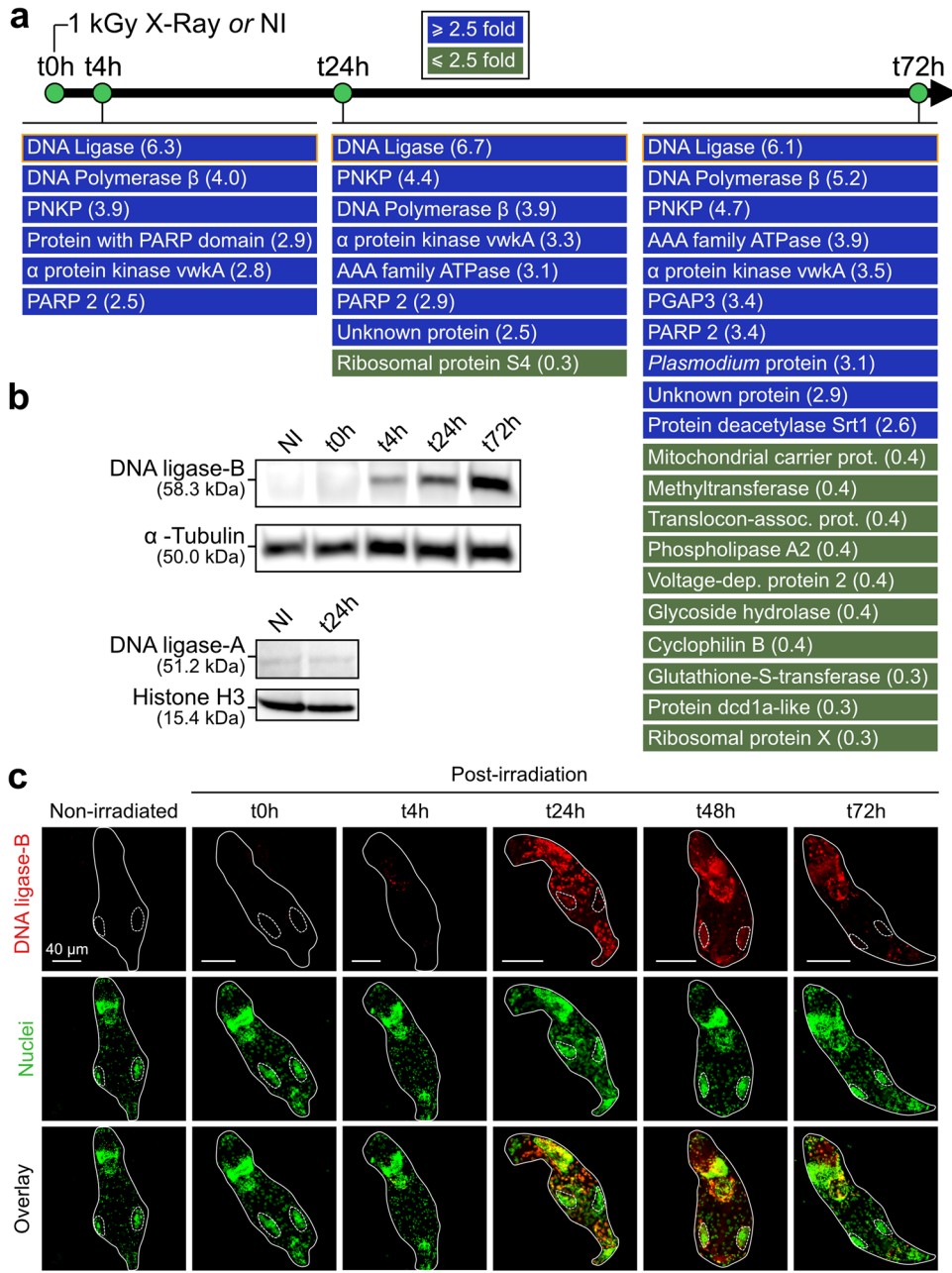

**Fig. 1 | Upregulation of a DNA ligase upon irradiation of the bdelloid rotifer**
***Adineta vaga* at 1 kGy of X-ray. a** Global proteomic analysis. Total proteins were
extracted from around 100,000 rotifers at t0h, t4h, t24h, and t72h post-irradiation.
Differential expression analysis of each protein was performed with Peaks studio.
This panel shows the proteins that are either over-represented (≥2.5x, blue) or
down-represented (≤2.5x, green) in comparison to the non-irradiated control (NI)
(see also Supplementary Table S1). **b** Single qualitative Western-blot analysis
showing the expression pattern of the DNA ligase copies A and B upon irradiation.

**c** Subcellular localization analysis of the induced DNA ligase (copy B) by immuno-
fluorescence at different timepoints post-irradiation. White lines delineate the
bdelloid rotifer *A. vaga* individual and dotted lines delimit the ovaries of *A. vaga* in
the middle part of their body, where less to no staining of DNA ligase-B was
observed. Nuclei were stained with DAPI but were displayed in green to ease the
interpretation of the result. A second replicate of this analysis is presented in
Supplementary Fig. S1d.

Analysis performed with our recently developed "Alienomics"
pipeline[25] strongly suggests that the IR-induced DNA ligase of *A. vaga*
is a horizontally transferred gene (Supplementary Table S2). The four
copies of the ligase gene have a slightly higher GC content than the *A.
vaga* genome average (35.1% vs 31.0%), their read coverage is the same
as the average coverage along the genome, and most importantly the
BLASTp analysis showed that the most closely related sequences
found in public databases do not belong to other metazoans but to
members of other eukaryote groups (i.e., Amoebozoa, fungi, Cilio-
phora and Chromista; Supplementary Table S2). Detailed sequence

analysis of the IR-induced DNA ligase of *A. vaga* shows that it is most
similar to the ligase E family of DNA ligases characterized by the
presence of two functional domains, the ubiquitous ATP-dependent
DNA ligase domain (pfam PF01068) and a specific OB-fold domain
(pfam PF14743) (Fig. 2)[34]. Phylogenetic analyses revealed that over
half of the members of this family are found in bacteria (>50% in Fig. 2
and >69% in Supplementary Fig. S2A), but other representatives were
also identified in several non-metazoan eukaryote groups and in few
animals (Fig. 2 and Supplementary Fig. S2A). In the phylogeny we
observe that both homoeologous pairs A and B of *A. vaga*, named

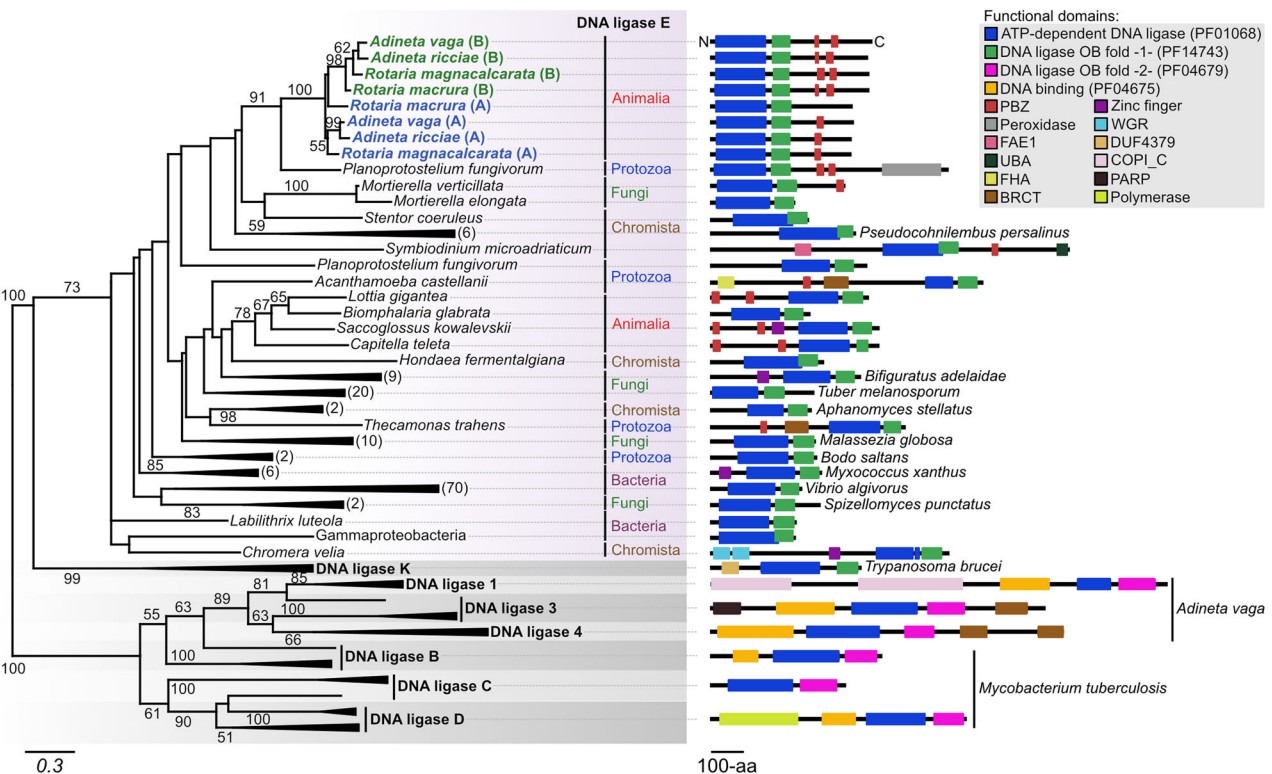

**Fig. 2 | Phylogeny of the induced DNA ligase in *Adineta vaga*, deriving from DNA ligase E.** The sequence of the DNA ligase over-represented in *A. vaga* following exposure to X-ray was compared to the available bdelloid rotifer sequences and to the uniref50 database using blastp (evalue 1e-05). The best 155 hits were extracted to perform an alignment using mafft and a phylogenetic analysis using iq-tree. Representative sequences of DNA ligases K, 1, 3, 4, B, C, and D were added separately from the initial blast search. The functional domains of the proteins were identified using interproscan and the structures were manually drawn for each representative case. Only boostrap values > 50 are indicated on the tree. Numbers between brackets represent the number of species of the same kingdom that are collapsed at each branch.

AvLigE-A and -B, are present in the genome of other bdelloid species, including the desiccation-sensitive species *Rotaria magnacalcarata* and *Rotaria macrura* (Fig. 2)[25,28,29]. However, this specific ligase is absent from the sequenced genome of the monogonont species *Brachionus plicatilis*, a rotifer clade closely related to bdelloid rotifers. These results suggest an ancient horizontal acquisition of this DNA ligase within the clade Bdelloidea, prior to the ancestral tetra-ploidisation of the genome, and being maintained both in desiccation resistant and sensitive bdelloid species. The exact origin of this ligase E gene in bdelloid rotifers is uncertain, its closest relative appears to be *Planoprotostellium* in all phylogenies obtained (Fig. 2 and Supplementary Fig. S2).

DNA ligases E form an evolutionary distinct lineage from the eukaryotic DNA ligase 1, 3 and 4 families and the prokaryotic DNA ligase B, C, and D families[34,35] (Fig. 2). Members of DNA ligase 1, 3, and 4 families were also found in the *A. vaga* genome, but only AvLigE-A and -B were identified as horizontally acquired and only AvLigE-B responded to IR in our proteomic analysis. DNA ligase K sequences from kinetoplastids were also added to the phylogenetic analysis because AvLigE was initially proposed to belong to this group of ligase K[30]. The structural organization of ligase K (including the presence of the two core domains pfam PF01068 and PF14743) indeed suggests that it is related to ligase E, but it is more divergent from AvLigE-A and B than other members of the ligase E family. The lack of taxonomic overlap between ligases K and E suggests that DNA ligase K might actually represent the DNA ligase E of kineto-plastids, which evolved more quickly within this lineage. This results in a higher molecular divergence which could produce inaccuracies during phylogenetic inference (i.e., long-branch attraction artifact). Note that in the complete phylogenetic tree with 502 sequences

(Supplementary Fig. S2A), DNA ligase K of kinetoplastids are placed within the main ligase E lineage making it difficult to conclude whether DNA ligase K and E correspond to two distinct ligase groups. Domain-specific phylogenetic trees of ATP-dependent DNA ligase, OB-fold, and both domains of Ligase E taken together were congruent with the phylogenetic tree obtained with the full-length ligase sequences (albeit with lower amount of phylogenetic signal), showing that both domains share the same evolutionary origin and history (Supplementary Fig. S2B).

In addition to the core functional domains, AvLigE proteins contain additional poly(ADP-ribose)(PAR)-binding zinc fingers (PBZ) domains at the C-terminus which are not present in the bacterial and in most other eukaryotic proteins (except *Thecamonas trahens* and *Mortierella verticillata*), although three other animal species do contain PBZ domains at the N-terminus of their ligase E, suggesting a later acquisition during metazoan evolution (Fig. 2). PBZ motifs are typically found in metazoans, and the majority of PBZ-containing proteins are involved in DDR, such as the nuclease APLF and SNM1 and the cell-cycle checkpoint protein CHFR, where they play a role in the recruit-ment of other proteins to DNA damage sites[36–40]. AvLigE-B has two PBZ domains whereas AvLigE-A has only one, and this is consistent for all ligase E genes of bdelloid rotifers (Fig. 2). This structural variation might be related to different activities of the proteins, AvLigE-B being strongly induced upon desiccation or radiation stress in *A. vaga* but AvLigE-A not or much less[30]. The PBZ domains of both AvLigE-A and AvLigE-B are surrounded by proline-serine-threonine (PST)-rich repeats. Such PST-repeats have recently been shown to promote H2AX-independent interaction between DNA and the DNA repair protein MDC1, serving as docking platform for other repair proteins at the site of DNA damage[41].

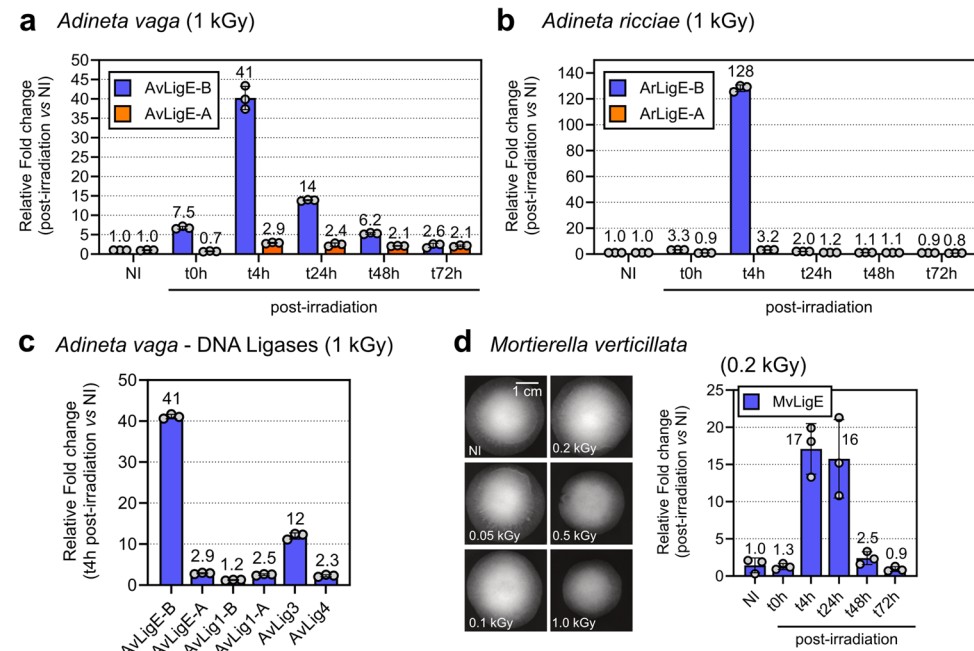

**Fig. 3 | Relative mRNA expression analysis of DNA ligases upon irradiation in bdelloid species *Adineta vaga* and *Adineta ricciae* and in the fungus species *Mortierella verticillata*. a, b, d** Kinetics expression analysis of DNA ligase E from *A. vaga*, *A. ricciae* and *M. verticillata*, respectively. The bdelloid rotifer species were irradiated at 1 kGy and *M. verticillata* at 0.2 kGy of X-ray. The mRNA level at t0h, t4h, t24h, t48h and t72h post-irradiation was determined by qPCR in triplicate. The relative expression were measured in comparison to the non-irradiated control (NI). In **d** the IR survival rate of *M. verticillata* was determined by its growth efficiency after X-ray irradiation to doses ranging from 0.05 to 1 kGy. A significant growth reduction was observed for radiation >0.2 kGy. *M. verticillata* has only one copy of DNA ligase E (MvLigE). **c** Relative expression analysis of AvLigE in comparison to the other DNA ligases of *A. vaga*. The rotifers were irradiated at 1 kGy and the mRNA levels were measured at t4h post-irradiation. Indicated fold changes correspond to the level of expression of each mRNA in comparison to the level of the same mRNA in the non-irradiated condition (NI). AvLig3 and AvLig4 have only one copy within the genome of *A. vaga*. The histograms represent the average values and standard deviations of three technical replicates.

## DNA ligase E is also over-represented in other bdelloid rotifer species and in the fungus *Mortierella verticillata* upon irradiation

Phylogenetic analysis highlighted that the AvLigE-A and -B have orthologs in other bdelloid rotifer species. Here we measured the relative mRNA expression of ligase E genes following radiation in the bdelloid rotifer *Adineta ricciae* in comparison to *A. vaga*. As both *Adineta* species can tolerate high levels of irradiation, they were exposed to 1 kGy of X-ray and total RNA was extracted at different timepoints post-irradiation (Fig. 3a, b). Only AvLigE-B in both *Adineta* species was induced upon irradiation with a strong expression at t4h (Fig. 3a, b), confirming the proteomic results on *A. vaga* (Figs. 1 and 3a, b). In desiccated *A. vaga* individuals an increase of transcript abundance of both copies A and B was detected, with B increasing twice as much as A[30], while after irradiation we barely detected copy A of ligase E in both bdelloid species tested and only copy B was strongly expressed.

To further determine the functional importance of AvLigE-B in comparison to the other DNA ligases of *A. vaga* upon DNA damage induction, we measured their relative mRNA expressions at t4h following 1 kGy of X-ray irradiation (Fig. 3c). Consistent with the proteomic analysis, the result showed that the AvLigE-B mRNA was induced up to 41-fold upon IR treatment, while expression of other ligases, including AvLigE-A, only showed a modest increase (Fig. 3a, d). A 12-fold induction was nevertheless observed for AvLig3, which is usually considered as the canonical DNA ligase involved in short-patch BER and SSB repair (Fig. 3d)[42]. These results further highlight the major role of AvLigE-B, probably during DNA damage repair in *A. vaga* and other bdelloid rotifers. We cannot rule out although the possibility that AvLigE-A and/or any other *A. vaga* DNA ligases might contribute to a basal level of cellular DNA ligase activity.

In our phylogenetic analysis we detected that the fungus species of the genus *Mortierella* contains a single copy of a DNA ligase E homolog (MvLigE) with a C-terminal PBZ-domain as observed in bdelloid species (Fig. 2). To determine whether this horizontally acquired ligase may have a similar adaptive role as the one proposed for AvLigE, we decided to study the radiation tolerance of *M. verticillata* and investigate whether MvLigE is also induced upon radiation exposure. Because the radio-tolerance of *M. verticillata* was never reported in the literature, we performed a survival assay and showed that the survival of the fungi was not affected up to 0.2 kGy of IR, with a progressive growth defect being observed at 0.5 and 1 kGy of X-ray radiation (Fig. 3d). The mRNA expression level of MvLigE was measured at the same timepoints (i.e., t0h, t4h, t24h, and t72h) after an exposure to 0.2 kGy of X-ray and the induction pattern was reminiscent to that observed with bdelloid rotifer ligases, showing a peak of expression at t4h and t24h post-irradiation (Fig. 3d) suggesting a plausible role of this actor in DNA repair of this fungus species.

## AvLigE-B is a functional DNA ligase

Even though AvLigE-B is over-represented upon DNA damage induction, the exact function of this protein is not known. The presence and strong induction of this DNA ligase is particularly intriguing since bdelloid rotifers possess all canonical eukaryotic DNA ligases normally involved in DNA replication and repair (see Fig. 3c). To assess the functionality of AvLigE-B, DNA ligation activity was measured in whole protein extracts from irradiated and non-irradiated *A. vaga* before and after depletion of the protein using a specific antibody (Fig. 4a). Incubation of irradiated protein extracts with magnetic beads coated with the anti-AvLigE-B antibody showed that the protein was specifically captured by the beads (Fig. 4b, lane 8), leading to an efficient depletion of AvLigE-B within the protein extract (Fig. 4b, lane 7).

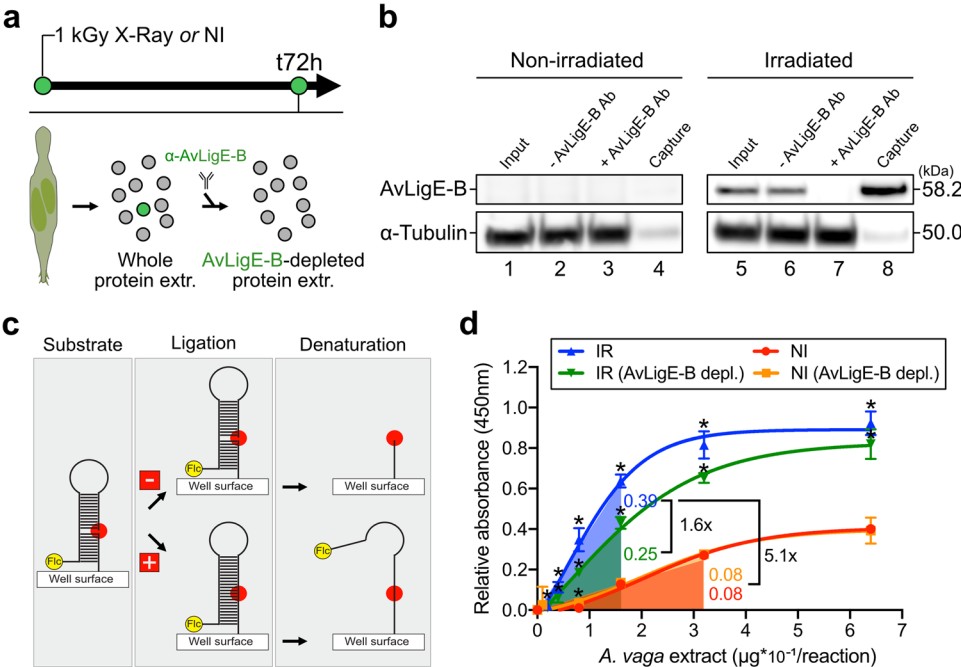

**Fig. 4 | In vitro DNA ligation assay to assess the functionality of AvLigE protein.** **a** Graphical scheme representing the workflow of immuno-depletion of AvLigE from *A. vaga* protein extracts. The rotifers were irradiated at 1 kGy to stimulate the production of AvLigE-B and the proteins were extracted at t72h post-irradiation. After extraction, AvLigE-B was immuno-depleted from the extracts. A non-irradiated control was also added to the analysis (NI). **b** Single qualitative Western-blot analysis demonstrating the efficiency of AvLigE-B immuno-depletion. See text for details. **c** Principle of the DNA ligase assay used. A complex of two oligonucleotides is assembled to form a hairpin loop containing a DNA nick to mimic a single-strand break (SSB). A fluorescein moiety is incorporated at the opposite end of the oligonucleotide structure. The complex is then incubated in the presence of increasing concentrations of *A. vaga* protein extracts, followed by a denaturation of the oligonucleotide structure. In the absence of any ligation activity, the oligonucleotide carrying fluorescein is lost from the wells during the denaturation process. Conversely, in the presence of a ligation activity, the DNA nick is sealed and the fluorescein moiety is kept in the wells after denaturation. The enzyme activity is determined by quantifying the amount of fluorescein retained in the wells as a

result of the denaturation step. Figure adapted from Healing et al. [75]. Flc, fluorescein. **d** DNA ligation activities with increasing concentrations of the various *A. vaga* protein extracts. Data represent the mean ± SD of four technical replicates. Differential activities were measured by comparison of the slopes of the relative absorbance at 450 nm in the linear portions of the curves. Statistical analysis in comparison to NI condition: one-sided Multiple unpaired t test, *$p$ value ≤ 0.05. Exact pvalues for NI-AvLigEB-depl. condition at concentrations from 0.0 to 6.4 $\mu$g*$10^{-1}$/reaction: >0.999999, 0.870033, 0.889157, 0.511038, 0.001045, 0.962556, 0.977686, 0.977686. Exact $p$ values for IR condition at concentrations from 0.0 to 6.4 $\mu$g*$10^{-1}$/reaction: 0.750223, 0.113340, 0.000260, 0.000260, 0.000114, 0.000001, 0.000027, 0.000016. Exact pvalues for IR-AvLigEB-depl. condition at concentrations from 0.0 to 6.4 $\mu$g*$10^{-1}$/reaction: >0.999999, 0.572836, 0.159447, 0.003717, <0.000001, 0.000012, 0.000002, 0.000154. IR, protein extract from irradiated *A. vaga* rotifers; IR (AvLigE-B depl.), protein extract from irradiated *A. vaga* rotifers and immuno-depleted for AvLigE-B; NI, protein extract from non-irradiated *A. vaga* rotifers; NI (AvLigE-B depl.), protein extract from non-irradiated *A. vaga* rotifers and immuno-depleted for AvLigE-B.

For the DNA ligation assay, a fluorescently-labeled SSB-containing DNA substrate was incubated with increasing amounts of protein extracts (Fig. 4c). Repair of the SSB generates a fluorescently-labeled ligation product, whose accumulation in the reaction can be measured by colorimetry (Fig. 4c). Analysis of the ligation activity of *A. vaga* protein extracts showed that irradiated *A. vaga* individuals significantly enhance (>5.1×) the global DNA ligation activity within the protein extract compared to a non-irradiated *A. vaga* extract (Fig. 4d, compare IR to NI). This suggests that DNA ligation, and particularly DNA SSB repair, is a fundamental process activated by bdelloid rotifers upon irradiation. Depletion of AvLigE-B from the protein extract of irradiated *A. vaga* decreased the ligation activity by 1.6-fold, demonstrating that AvLigE-B is an active DNA ligase (Fig. 4d, compare IR to IR-AvLigE-B depl.). However, depletion of AvLigE-B did not reach the ligation activity seen in the non-irradiated condition (Fig. 4d, compare NI to IR-AvLigE-B depl.). This suggests that AvLigE-B is not the only ligation proficient enzyme that is induced upon irradiation, even though it contributes to about 35% of the ligation activity measured in stressed cells.

### Heterologous expression of AvLigE improves radio-tolerance of human cells

Our study reveals that the horizontal acquisition of a non-metazoan DNA ligase likely contributes to the high DNA damage tolerance in

bdelloid rotifers. We next sought to find out whether heterologous expression of AvLigE in human cells might also improve their DNA damage tolerance. AvLigE-A and AvLigE-B genes were stably trans-fected using CRISPR-cas9 system and constitutively expressed in human HEK 293T cells (Fig. 5a). RT-qPCR confirmed that both genes were expressed at a similar level (1:1.1 ratio of relative mRNA concentration for AvLigE-A *vs* AvLigE-B) and the expression of both ligases was shown to significantly increase the DNA ligation activity in HEK 293T cell extracts, albeit at a higher rate with AvLigE-A than with AvLigE-B (i.e., 5.4- vs 1.9-fold respectively) (Fig. 5b).

Strikingly, the measured increase in ligation activity was found to correlate to the rate of cell survival upon IR exposure, as measured by colony formation assay (Fig. 5c). At 0.002 kGy of IR, the cell line expressing AvLigE-A (but not AvLigE-B) showed a 1.4 fold increase of survival rate compared to the non-irradiated cell line, while at 0.004 KGy, expression of both AvLigE-B and AvLigE-A significantly improved cell survival by 1.7 and 2.8 fold, respectively (Fig. 5c). This increased proliferation potential is further supported by morphological analysis showing that human cell colonies expressing AvLigE-A had a normal size and shape under both irradiation conditions, whereas the control HEK 293T cell line (and to some extent, the AvLigE-B expressing cells) formed colonies with strongly altered morphology likely due to more frequent cell cycle arrest, apoptosis and necrosis (Fig. 5d).

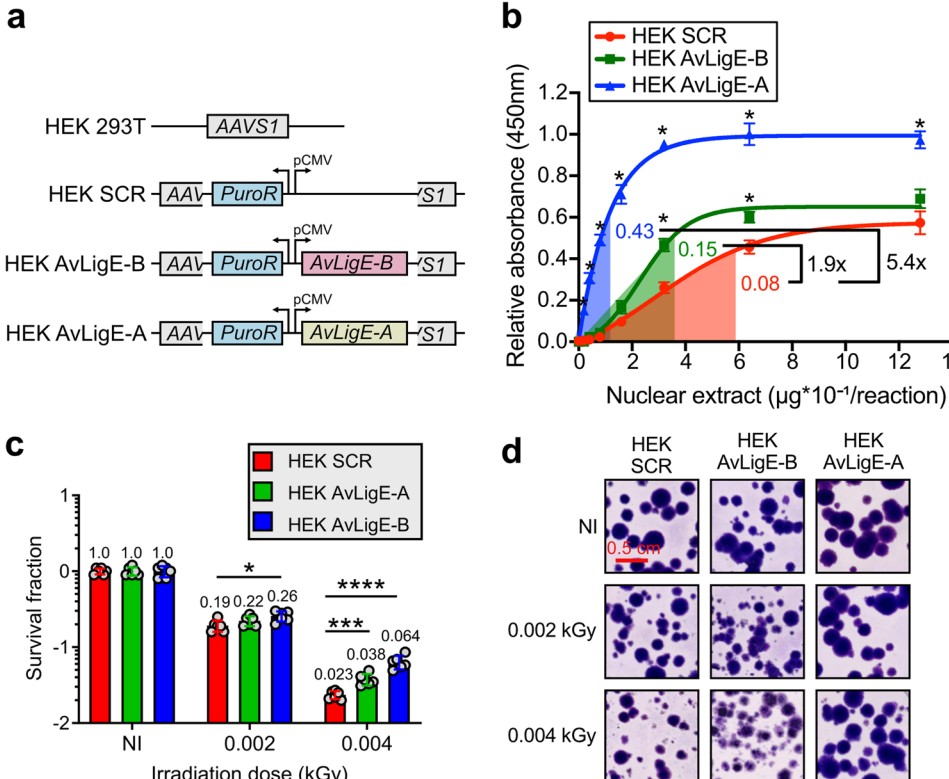

**Fig. 5 | AvLigE improves X-ray tolerance of human cell lines. a** Graphical scheme describing the HEK 293T derivative cell lines expressing the AvLigE variants. The control cell line (HEK SCR) includes the empty pDNR plasmid. The other two cell lines express either *AvLigE-B* gene or *AvLigE-A* gene under the constitutive CMV promoter. All constructed cell lines are resistant to puromycin. The cell lines expressed (tested by RT-qPCR) both genes at a similar level: 1:1.1 ratio of relative mRNA concentration for AvLigE-A *vs* AvLigE-B. **b** In vitro DNA ligation assay (same assay as in Fig. 4c) with the different variants of the HEK 293T cell line. Differential activities were measured by doing a ratio of the slopes in the linear portions of the activity curves. Data represent the mean ± SD of four technical replicates (*n* = 4). Statistical analysis in comparison to HEK SCR condition: one-sided Multiple unpaired t test, *p value ≤ 0.05. Exact pvalues for HEK-AvLigE-B at concentrations from 0.0 to 12.8 μg*10⁻¹/reaction: 0.373901, 0.218750, 0.078090, 0.076832, 0.091878, 0.007156, 0.027471, 0.136681. Exact *p* values for HEK-AvLigE-A at concentrations from 0.0 to 12.8 μg*10⁻¹/reaction: 0.373901, 0.000508, 0.000170,

0.000053, 0.000152, 0.000035, 0.000437, 0.001109. **c** Survival fraction (logarithmic scale) of the different human cell lines upon X-ray irradiation as assessed by colony formation assay. Cells were irradiated at 0.002 or 0.004 kGy. Survival was normalized to the fraction of surviving cells in the non-irradiated condition. Statistical analysis was performed to compare the survival fraction in comparison to the HEK SCR control condition using one-sided multiple *t*-tests. Significant values: *p value ≤ 0.05; ***p value ≤ 0.001; ****p value ≤ 0.0001. Exact pvalues for HEK-AvLigE-B condition: >0.999999 in NI condition, 0.273588 at 0.02-kGy, and 0.001981 at 0.04-kGy. Exact pvalues for HEK-AvLigE-A condition: >0.999999 in NI condition, 0.022496 at 0.02-kGy, and 0.000071 at 0.04-kGy. Data represent the mean ± SD of two biological replicates, each including three technical replicates (*n* = 6). NI non-irradiated control. **d** Representative images of the colonies formed by the variable human cell lines in the different conditions of irradiation. NI non-irradiated control.

## Discussion

In this study, we identify and characterize the exogenous acquisition of a DNA ligase belonging to the ligase E family of prokaryotic ligases within bdelloid species, being strongly induced upon irradiation in two distinct *Adineta* species. A homology-based analysis showed that AvLigE is found in all sequenced bdelloid rotifer species while it is absent in their sister-group, the monogonont rotifers. The identification of four copies of the gene indicates that an ancestral LigE copy was acquired before the tetraploidization and radiation of the bdelloid species[25,43], but likely after the split between them and monogononts. Phylogenetic and sequence analyses confirmed the non-metazoan origin of this gene, with the catalytic core (ATP-dependent DNA ligase and OB-fold domains) being similar to the prokaryotic DNA ligase E. Whereas prokaryotes mainly depend on NAD⁺-dependent DNA ligases, ATP-dependent ligases E are the second most represented ligases in bacteria, being found in Proteobacteria, Cyanobacteria, and Firmicutes[34]. DNA ligases E can act on different DNA substrates, including nicks, cohesive ends, and mismatches[44], being more versatile than the canonical metazoan DNA ligases. Therefore, bdelloid rotifers may have maintained this horizontally acquired gene to deal with a variety of DNA damages directly arising from genotoxic stresses like

desiccation, freezing or environmental radiations, or from the secondary production of ROS encountered in their semi-terrestrial habitats.

Phylogenetic analyses also revealed that the closest orthologous ligase E sequences of bdelloid rotifers are found in the groups of Protozoa, Fungi, and Chromista (Fig. 2). This may suggest a horizontal transfer of Ligase E from an eukaryotic non-metazoan group to an ancestor of bdelloid rotifers, while similar horizontal transfers of ligase E might also have occurred independently in different phyla to adapt to extreme genotoxic stresses. In order to investigate this, we studied the expression of ligase E in the fungus species *Mortierella verticillata* exposed to X-ray radiation. As for *A. vaga* and *A. ricciae*, *M. verticillata* survived doses above 0.2 kGy and its single copy of DNA ligase E was induced upon DNA damages following exposure to X-ray radiation, suggesting a plausible convergent role of Ligase E in distinct eukaryotic clades. Bdelloid rotifers and the soil fungus *Mortierella verticillata* inhabit semi-terrestrial environments where they frequently experience complete dryness, sometimes during prolonged periods, inducing oxidative stress and DNA damages. Organisms prone to accumulate DNA DSBs during desiccation, such as bdelloid rotifers[13], could potentially have more opportunities for horizontal gene

transfer[45,46]. The ecological proximity of bdelloid rotifers with bacteria, fungi and protozoans in habitats such as lichens or soil, also provides an opportunity for uptake of this non-metazoan DNA. Moreover, genes providing a direct selective advantage are more amenable to become effective HGT, such as this ligase E. Therefore, our study demonstrates that horizontal gene transfer in eukaryotes can play a major role in driving adaptation, with bdelloid rotifers being a key model system having the highest number of foreign genes among published animal genomes[25]. Recently, Rodriguez et al. identified in bdelloid rotifers a specific gene coding for a N4-cytosine methyl transferase (N4CMT) that was acquired from bacteria and efficiently integrated in the epigenetic regulation system of *A. vaga* to silence its transposons[47], further illustrating the importance of horizontally acquired genes in the adaptive evolution of bdelloid rotifers. It is also interesting to note that other copies of DNA ligase E were detected in a few animals *Saccoglossus kowalevskii*, *Lottia gigantea*, *Biomphalaria glabrata*, and *Capitella teleta* (see Fig. 2). Even though the degree of conservation and the potential activity of this DNA ligase within these animal species require more investigation, its detection within these organisms living in marine coastal or shallow-water ecosystems that are stressful environments subjected to drastic redox imbalance, might suggest a possible role of this DNA ligase in DDR induced by these stresses[48,49].

When compared to bacterial DNA ligase E, the horizontally acquired LigE in bdelloid rotifers contains additional eukaryotic PBZ domains known to be involved in interaction with other DNA repair proteins[40]. These PBZ domains are surrounded by proline-serine-threonine (PST)-rich repeats that are known to promote chromatin interaction for the DNA damage regulator MDC1[41]. Interestingly, this interaction between MDC1 and the chromatin was also shown to mediate the recruitment of other repair proteins independently of H2AX[41]. Since H2AX is absent in *A. vaga*[50], PST-containing proteins like AvLigE might play a dual role by directly interacting with the DNA to repair DNA lesions, but also recruiting other DNA repair proteins at the site of damage through its PBZ domains, in the absence of H2AX signaling pathway.

In our comparative proteomic analysis we found that the majority of over-represented proteins following *A. vaga* irradiation are involved in DNA repair. In particular, DNA polymerase $\beta$, the polynucleotide kinase/phosphatase PNKP, PARP proteins and DNA ligases are main actors of the BER pathway which notably corrects base lesions that are generated by ROS during the X-ray-induced radiolysis of water molecules[32]. Despite the strong oxidizing environment following IR, none of the many antioxidant repertoire genes of *A. vaga*[25,28], nor other previously characterized stress proteins or chaperones, were found to be significantly induced upon X-ray radiation in our analysis. This finding suggests that the oxidative stress resilience of bdelloid rotifers largely relies on the constitutive expression of antioxidant genes, as observed in the transcriptomic analyses following desiccation[30]. In tardigrades, little variation was also observed in the level of transcripts coding for antioxidant enzymes during dehydration and rehydration[19]. Constitutive expression of antioxidant genes appears to be specific of bdelloid rotifers and tardigrades since in other eukaryotic cells, their expression seems tightly regulated to ensure redox homeostasis (e.g., see refs. 51,52). It has been hypothesized that the constitutive expression of an antioxidant defense allows bdelloids and tardigrades to be prepared to frequent and unpredictable environmental changes in their semi-terrestrial habitats and to keep the DNA repair proteins protected from oxidative damage[11]. The resilience to oxidative stress in bdelloid rotifer *A. vaga* is being further investigated.

The functionality of AvLigE was assessed by showing that the depletion of the ligase from *A. vaga* protein extracts leads to a reduction of in vitro SSB repair. However, we provided further evidence regarding the potential activity of AvLigE in DNA repair by showing that its expression in human cell lines also improved their survival to X-ray radiation, which also demonstrates how broadly an

exogenous acquisition of a ligase E-derivative could improve radiotolerance. This strategy of over-expression of DNA repair proteins in human cells has been previously explored to enhance the efficiency of endogenous DNA repair[53]. This approach has not always been successful, with some proteins clearly improving DNA repair potential of the cells (e.g., over-expression of DNA ligase III in mitochondria[54]) while others resulting in increased genome instability (e.g., over-expression of DNA polymerase $\beta$[55,56]). Our experiment showed that the expression of the exogenous AvLigE was significantly enhancing radiotolerance of human cells upon radiation, with a higher effect of copy A of AvLigE. This finding suggests that this copy of the protein was efficiently integrated within specific pathways of human cells, probably involved in DNA repair, without affecting the physiology of the cell. To the contrary, in the proteomic analysis done in vivo on *A. vaga* bdelloid rotifer, only copy B of the gene was identified upon X-ray radiation, while this was not the major contributor of enhanced radiotolerance in human cells. This difference between the ligation activity of AvLigE-A and -B in human cells vs bdelloid rotifers is likely due to a difference in stability or functionality of the proteins within one system or the other and requires further investigations. Nevertheless, this observation of a functional divergence between AvLigE-A and AvLigE-B suggests a divergent transcriptional regulation of the two homoeologous genes, which is frequent in polyploid species like plants (see e.g. refs. 57–61).

It is tempting to speculate that bdelloid rotifers maintained this DNA ligase E gene in addition to the standard DNA ligases to specifically fulfill the function of DNA repair of numerous DNA damages. The evolutionary potential of having multiple DNA ligases presumably lies in the ability to separate the work between different enzymes, each acting in distinct pathways on specific substrates (i.e., DNA replication, repair of base lesions, repair of DSBs, etc.)[42], or to have an additional versatile DNA repair protein that can act on many types of damages, likely improving the repair efficiency. Increasing the number of DNA repair proteins might also promote DNA repair rather than cell death in front of DNA damaging conditions, as it has been described in polyploid tumor cells for which the over-expression of DNA repair proteins results in the bypass of pathways leading to senescence or apoptosis[62]. Interestingly, the *p53* gene, a critical regulator of apoptosis, is absent from bdelloid rotifer genomes. Moreover, these organisms are eutelic, having no mitotic divisions of their somatic cells at adult stage. Following massive DNA fragmentation after exposure to high doses of IR, the DNA in the somatic nuclei of *A. vaga* gets repaired within 24 h after recovery, producing a partially reassembled genome[13,18]. Therefore, like for cancer cells (which often contain a *p53* mutation[63]), even under harsh DNA damaging conditions, bdelloid rotifers do not activate a controlled cell death pathway but rather repair the damages incurred. The AvLigE detected here is likely involved in this rapid repair of somatic DNA damage, being over-represented 4 h after IR in our proteomic analysis and showing a nuclear localization in somatic nuclei of *A. vaga* at later timepoints (Fig. 1). Somatic nuclei are indeed the most abundant nuclei within a rotifer individual, providing the strongest signal in the proteomic analysis. In addition, Terwagne et al. [18] showed that the repair of germ line genomic lesions induced by ionizing radiation in *A. vaga* is delayed to a specific time window of oogenesis during which homologous chromosomes adopt a meiotic-like juxtaposed configuration, resulting in a complete reconstitution of their genome. Bdelloid rotifers therefore appear to have optimized their DNA repair mechanisms towards a rapid response of G0/G1 somatic nuclei likely involving DNA ligation, as in the non-homologous end joining (NHEJ) and BER pathways, while the G2 germline cells of *A. vaga* rather involve a homology-directed DNA repair during a modified meiosis to accurately repair their DNA in the maturing oocyte[18].

Our results provide new evidence of the key role of HGTs in adaptive evolution of eukaryotes. The horizontally acquired DNA ligase E seems to improve DNA damage tolerance of bdelloid rotifers,

which may have favored the exploitation of new environmental niches and contributed to their evolution[27,64]. Even though the importance of lateral gene transfers in the acquisition of new functions is presumably not limited to bdelloid rotifers, these studies demonstrate that these organisms are interesting model systems to decipher the molecular and cellular mechanisms underlying these processes. It also demonstrates that horizontal gene transfer can integrate specific regulatory processes in metazoans driving evolutionary innovations and opening the way to potential applications.

## Methods

### Bdelloid rotifer culture

All experiments were performed using isogenic *A. vaga* clones derived from a single individual from Meselson Laboratory[50]. The cultures were maintained hydrated in cell flasks with natural spring water (Spa®), at 25 °C and fed with lettuce filtrate.

### Comparative proteomic analysis upon X-ray irradiation of *A. vaga*

Each condition of irradiation was performed in duplicate with about 100,000 hydrated *A. vaga* individuals starved overnight (O/N) in spring water including 1% of Penicillin/Streptomycin (ThermoFisher Scientific). X-ray radiations (0.8–1 kGy) were performed using a Precision X-ray X-RAD 225 XL instrument (225 kV, 19 mA, filter 1, 30 cm). After radiation, the rotifers were incubated at 25 °C until the extraction time. The proteins were extracted at t0h, t4h, t24h, and t72h post-irradiation.

For protein extraction, rotifers were collected from the plate and centrifuged at 4000 rpm, 4 °C, 10 min. They were then washed twice in 10 ml of ice-cold PBS (centrifugation 4000 rpm, 4 °C, 10 min). After the last wash, the supernatant was removed and the pellet was resuspended in 300 μl of cold lysis buffer (10 mM Tris, pH 8.0; 1 mM EDTA, pH 8.0; 5 mM DTT; 1% CHAPS, 1 M KCl; 1 mM PMSF, 25% of glycerol and protease inhibitors). The resuspended pellet was then transferred to a cold Dounce and the rotifers were lyzed by 1000 Dounce pulses with tight pestle B to crack the cuticle of the animals. The lysate was then recovered from the Dounce in an Eppendorf tube and stored O/N at −80 °C. The cellular debris and insoluble proteins were then pelleted by ultracentrifugation at 42,000 rpm, 3 h, 4 °C. Supernatant containing soluble proteins was dialyzed against the dialysis buffer (20 mM Tris, pH 8.0; 20% glycerol; 100 mM KoAc; 0.5 mM EDTA, pH 8.0; 1 mM DTT) in Slide-A-Lyzer™ MINI Dialysis Device, 3.5 K MWCO, 2 mL (ThermoFisher Scientific) according to the manufacturer protocol. After O/N dialysis at 4 °C, the extracted proteins were recovered, the concentration was determined using Pierce 660 nm Protein Assay Reagent (ThermoFisher Scientific), and the proteins were flash frozen in liquid nitrogen before storage at −80 °C.

Filter-aided sample preparation (FASP) was used to generate tryptic peptides from protein mixtures prior to mass spectral analysis. The filter of Microcon-30 kDa (Millipore) was first rinsed with 100 μl of 1% Formic acid (v/v) and centrifuged 15 min at 14,500 rpm. Fifty μg of protein extract were brought to 150 μl using Buffer UA (8 M urea in 0.1 M Tris/HCl pH 8.5). The extract was then added to the column and centrifuged 15 min at 14,500 rpm. The filter was then washed three times with 150 μl of Buffer UA (15 min at 14,500 rpm). For the reduction step, the samples were treated with 100 μl of DTT, incubated on a thermomixer at 24 °C for 15 min, and centrifuged 10 min at 14,500 rpm. Filter was further rinsed with 100 μl of Buffer UA (15 min at 14,500 rpm). For the alkylation step, 100 μl of IAA (0.05 M iodoacetamide in UA) was added on the filter and incubated 20 min at 24 °C in the dark before centrifugation 10 min at 14,500 rpm. Filter was then rinsed with 100 μl of Buffer UA (15 min at 14,500 rpm). The IAA was quenched by the addition of 100 μl of DTT. After an incubation of 15 min at 24 °C, the column was centrifuged 10 min at 14,500 rpm, and further rinsed with 100 μl of Buffer UA. After centrifugation 10 min at

14,500 rpm, the samples were treated three times with 100 μl of Buffer ABC (0.05 M $NH_4HCO_3$ in water) and centrifuged 10 min at 14,500 rpm. For the digestion step, 1 μg of trypsin (Promega) diluted in Buffer ABC was added to the column and incubated O/N at 24 °C on a thermomixer under agitation (300 rpm). The digested proteins were recovered by placing the Microcon column on a protein LoBind tube of 1.5 ml followed by centrifugation 10 min at 14,500 rpm. The filter was rinsed with 40 μl of Buffer ABC and centrifuged 10 min at 14,500 rpm. The filtrate was then acidified with a solution of 10% Trifluoroacetic acid (TFA) to reach 0.2% TFA final concentration. The sample was finally dried in a speed-vac and the pellet was resuspended in 20 μl of 2% acetonitrile (ACN) and 0.1% TFA before mass spectrometry analysis.

The digested proteins were analyzed using nano-LC-ESI-MS/MS tims TOF Pro (Bruker) coupled with a UHPLC nanoElute (Bruker). Peptides were separated by nanoUHPLC (nanoElute, Bruker) on a 75 μm ID, 25 cm C18 column with integrated CaptiveSpray insert (Ionopticks) at a flow rate of 400 nl/min, at 50 °C. LC mobile phases A was water with 0.1% formic acid (v/v) and B was ACN with formic acid 0.1% (v/v). The sample was loaded directly on the analytical column at a constant pressure of 800 bar. The digest (1 μl) was injected, and the organic content of the mobile phase was increased linearly from 2% B to 15% in 18 min, from 15% B to 25% in 9 min, from 25% B to 37% in 3 min and from 37% B to 95% in 5 min. Data acquisition on the tims TOF Pro was performed using Hystar 5.0 and otof-Control 6.0. tims TOF Pro data were acquired using 160 ms TIMS accumulation time, mobility (1/K0) range from 0.7 to 1.4 Vs/cm². Mass-spectrometric analysis were carried out using the parallel accumulation serial fragmentation (PASEF) acquisition method. One MS spectra followed by six PASEF MSMS spectra per total cycle of 1.16 s. Two injections per sample were done.

Data analysis was performed using PEAKS Studio X+ with ion mobility module and Q module for label-free quantification(Bioinformatics Solutions Inc.). Protein identifications were conducted using PEAKS search engine with 15 ppm as parent mass error tolerance and 0.05 Da as fragment mass error tolerance. Carbamidomethylation was allowed as fixed modification, oxidation of methionine and acetylation (N-term) as variable modifications. Enzyme specificity was set to trypsin, and the maximum number of missed cleavages per peptide was set at 2. The peak lists were searched against the protein database from *Adineta vaga* (NCBI, assembly ASM2161353v1) containing 33,103 entries. Peptide spectrum matches and protein identifications were normalized to less than 1.0% false discovery rate. Label-free quantitation (LFQ) method was based on expectation - maximization algorithm on the extracted Ion chromatograms of the three most abundant unique peptides of a protein to calculate the area under the curve. For the label-free quantitation results, peptide quality score was set to be ≥4 and protein significance score threshold was set to 20. The significance score was calculated as the −10log10 of the significance testing *p*-value (0.01). ANOVA was used as the significance testing method. The minimum number of peptides from a protein to be identified per condition was set to 2 and only peptides present in at least half of the samples from the same condition were considered in the analysis. Total ion current was used to calculate the normalization factors. At a threshold of 2.5 the same proteins were consistently overrepresented at the different timepoints, highlighting the significance of the specific biological processes uncovered. The mass spectrometry proteomics data have been deposited to the ProteomeXchange Consortium via the PRIDE[65] partner repository with the dataset identifier PXD043051 and 10.6019/PXD043051. A summary of the data is provided in Supplementary Table S1.

### Genomic integrity analysis

Genomic integrity was assessed by pulsed-field gel electrophoresis (PFGE) as described in Hespeels et al. 2014[13].

## DNA ligase expression analysis by Western-blot

Custom polyclonal antibodies targeting the copy A (AvLigE-A) or copy B (AvLigE-B) of the DNA ligase were produced by a speedy 28-Day rabbit immunization programme (Eurogentec). Two rabbits were injected with the synthetic peptide C + NVVEKKPVRKRTKQS-cooh for the copy A and C + TTDNNEESNKRM KTD-cooh for the copy B using KLH as carrier protein. At the final bleed, the specific IgGs targeting the DNA ligase peptide were purified by affinity.

For the Western-blot, 10 μg of protein extract (see above for preparation of the extract) were separated on SDS-PAGE gel and then transferred to a PVDF membrane. The membrane was blocked for 1 h with 5% w/v non-fat dry milk diluted in TBS-Tween 0.1%. The anti-AvLigE-A or AvLigE-B antibodies were then added (1:200) in TBS-T buffer supplemented with 1% w/v non-fat milk for 1 h30 at RT. Afterwards, the membrane was washed three times in TBS-T and incubated with 1:1,000 anti-rabbit secondary antibody coupled to HRP (ThermoFisher Scientific) in TBS-T −1% milk for 1 h at RT. The membrane was then washed three times in TBS-T and finally revealed with Super Signal West Pico Chemiluminescent substrate (ThermoFisher Scientific). The chemiluminescent signal was captured with an Imager AI600 (Amersham).

For tubulin or Histone H3 detection, just after the revelation of the AvLigE-B signal, the same membrane was washed in TBS-T and incubated with 1:2500 commercial anti-tubulin antibody (Sigma Aldrich) or 1:5000 anti-histone H3 antibody (Bethyl Laboratories) in TBS-T −1% milk for 1 h at RT. The membrane was then incubated with 1:2500 anti-mouse secondary antibody coupled to HRP (ThermoFisher Scientific) in TBS-T − 1% milk for 1h at RT and revealed as described above.

For the detection of recombinant AvLigE-A or AvLigE-B expressed in E. coli, cells containing pET21a plasmids with 6xHis-tagged fusions of AvLigE-A or AvLigE-B were first selected. The expression of the proteins was induced by the addition of 1 mM IPTG during 24 h and the cells were then collected before lysis (10 mM Tris pH 7.4, 500 mM NaCl, 1 mM DTT, 1 % Triton X-100, 10% Glycerol, 1 mM PMSF, protease inhibitors). After lysis, the cells were sonicated at high intensity and the lysate was clarified by centrifugation at 4000 rpm for 40 min at 4 °C. About 10 μg of total protein from each condition were loaded on a denaturing SDS-PAGE gel and the proteins were transferred on a nitrocellulose membrane. Ponceau staining was used to control the loading of the proteins and the proteins were detected by incubation with 1:400 of anti-AvLigE-A or AvLigE-B or 1:5000 of anti-His tag antibody (ThermoFisher Scientific).

## Subcellular localization of the DNA ligase

The induced DNA ligase was localized by immuno-fluorescence within the bdelloid rotifer A. vaga following an irradiation at 1 kGy of X-rays. The rotifers were first collected and concentrated by centrifugation (4000 rpm, 4 °C, 10 min) at different time points post-irradiation - t0h, t4h, t24h, t48h, and t72h. The individuals were resuspended in 20 μl of spring water and spread on a SuperFrost Plus microscope slide (ThermoFisher Scientific). The external cuticle of the rotifers was then removed by freeze-cracking. For that, the organisms were covered by a coverslip, gently squashed, and drown in liquid nitrogen. Immediately after the freezing, the coverslip was removed using a razor blade, and the slide was left at RT for few minutes. The rotifers were then fixed within 1% Formaldehyde for 10 min at RT, followed by three washes in PBS. Blockage was realized by incubation of the slide within a solution of 1% BSA and 0.3 % Triton-X100 for 1 h at RT. The primary antibody (custom anti-AvLigE-B) was diluted at 1:150 in the blockage solution. A volume of 200 μl of that dilution was dropped on the slide, covered by a coverslip and incubated at 4 °C O/N. The primary antibody was then washed three times in PBS. The secondary antibody - Alexa 568 Goat anti-rabbit (ThermoFisher Scientific) - was diluted at 1:150 in the blockage solution and a volume of 200 μl was dropped on the slide, and incubated for 1 h at RT. After 3 washes in PBS, the nuclei were stained with a 1:1,000 DAPI solution diluted in PBS and incubated

20 min at RT. After three final washes in PBS, the slides were mounted with Mowiol and sealed with a coverslip. The images were captured with a Zeiss LSM 900 with Airyscan2 in Z-stacks with constant parameters. Mounting and analysis of the images were done with the Fiji software (v2.1.0/1.53c).

## DNA ligase gene evolutionary history analysis

To determine if the induced DNA ligase in A. vaga was acquired through horizontal gene transfer (HGT), the Alienomics pipeline was used to detect both HGTs and contaminants in the genome assembly of A. vaga[25]. Alienomics combines several genomic parameters such as gene taxonomy, GC content, sequencing depth, synteny and expression data, to detect HGTs (along with potential contaminants).

For the phylogenetic analysis, the AvLigE copy B protein sequence (UJR14855.1)[25] was used as query to perform a BLAST search using default parameters against protein data from other bdelloid species for which genome assemblies were available (i.e., Adineta ricciae, Rotaria magnacalcarata and Rotaria macrura)[28,29]. Bedtools function getfasta was then used to extract the matched sequence into a fasta file[66]. Sequences were then aligned using mafft[67] with default parameters and the resulting alignment was inspected in order to manually fuse back non-overlapping hits from a given protein. The aligned bdelloid sequences were kept for later use.

AvLigE-B sequence was also used to perform a diamond blastp search[68] against uniref50 database using the following parameters: −more-sensitive -k 20000 −max-hsps 1 −evalue 1e-05 −query-cover 20. A total of 502 matched subject sequences were retrieved from the Uniref50 database, out of which only the best 155 hits were kept for subsequent analysis. These sequences were added to the bdelloid sequences previously retrieved as well as to sequences corresponding to ligases 1, 3, 4, B, C, D and K retrieved from NCBI (see Supplementary Table S3 for a list of accession numbers used in this analysis). These steps resulted in the production of two datasets: one comprising 543 sequences (502 blast hits combined with bdelloid sequences and other ligases) and one comprising 184 sequences (155 best blast hits combined with bdelloid sequences and other ligases).

Both datasets were then aligned using mafft with default parameters[67]. A third dataset based on the 184 sequence alignment was produced by only retaining both ATP and OB ligase proteic domains from the full ligase sequences. For the three datasets, alignment columns for which more than 80% missing data was observed were discarded using custom perl script. Considering the 184 sequence alignments, this reduced its length from 4935 aa to 529 aa. The reduced alignment was used as input for a phylogenetic analysis under maximum likelihood using iq-tree 1.6.10[69]. The following parameters were used for the 184 sequence alignments: -b 100 -m C20 + R4. The following parameters were used for the three alignments: -b 100 -m LG + R4 + F. The resulting trees were visually inspected with Archaeopteryx[70] and iTol[71] with several lineages collapsed for clarity when needed. Complete sequence sets (i.e. raw, aligned and reduced) as well as the phylogenetic tree are available on Figshare: https://figshare.com/s/7cbed8c5d59abdd2f4ae.

Conserved protein domains were predicted in the different sequences from DNA ligases using InterPro[72] and illustrated with Affinity Designer. Only one representative sequence was illustrated when several sequences were collapsed. The kingdom classification for each organism was determined according to Ruggiero et al. (2015)[73].

## Quantitative PCR analysis

For the bdelloid rotifers A. vaga and A. ricciae, prior to irradiation, about 100,000 rotifers were starved 24 h in the presence of 1% Penicillin/Streptomycin (ThermoFisher Scientific). Rotifers were irradiated at 1 kGy of X-rays as described above.

For M. verticillata, fungi (ATCC 42662) were first grown on YM agar and then in liquid YM broth. The survival to X-ray radiation doses

of 0.05, 0.1, 0.2, 0.5 and 1.0 kGy was performed with an homogenous suspension of hyphae in liquid. After irradiation, an identical volume of liquid was plated on YM agar and incubated at 25 °C during 3 days. Survival potential was determined by doing a ratio of the diameter of the growing area in comparison to the non-irradiated condition. The experiment was performed in triplicate. The expression analysis of DNA ligase E from *M. verticillata* was performed after an irradiation at 0.2 kGy of X-rays.

Total RNA of rotifers or fungi was extracted at different times post-irradiation: t0h, t4h, t24h, t48h, and t72h. A non-irradiated control was also added to the experiment. Total RNA extracts were prepared in Trizol (ThermoFisher Scientific) according to the manufacturer's instructions. Any residual DNA was digested with DNase I (ThermoFisher Scientific) and total RNA was reverse transcribed using the SuperScipt III First-Strand Synthesis System for RT-PCR (ThermoFisher Scientific). Gene expression level was determined with Applied Biosystems 7500 instrument (ThermoFisher Scientific) using the SsoAdvanced Universal SYBR Green Supermix (Biorad). The reactions were performed in triplicate for each condition.

To compare the relative abundance of the DNA ligase transcripts between irradiated and non-irradiated conditions, the real-time quantitative PCR experiments were performed with primers targeting the *gapdh* gene (=housekeeping gene) or the specific DNA ligase gene of each organism. All primer designs were performed using the online ePrimer3 software using the standard settings. The oligonucleotides used in this study are listed in Supplementary Table S4. Relative gene expression data was determined using the Livak method[74].

### In vitro DNA ligation assay

The in vitro DNA ligation activity of *A. vaga* cell extracts was assessed by using an assay described by Healing et al. [75]. The *A. vaga* whole protein extracts were prepared as described for the proteomic analysis except that the dialysis was performed in the absence of DTT to avoid the reduction of the antibody light and heavy chains during the immuno-depletion. For the ligation assay with HEK 293T cell lines expressing AvLigE-A and AvLigE-B (see below), we prepared a nuclear protein extract using the NE-PER extraction kit (ThermoFisher Scientific) following the instructions from the manufacturer.

For the immuno-depletion of AvLigE, we used the Dynabeads Protein A Immunoprecipitation kit (ThermoFisher Scientific) following the instructions of the manufacturer. A quantity of 5 µg of AvLigE-B antibody was bound and crosslinked to the magnetic beads using the BS³ crosslinker. A total of 100 µg of each protein extract was incubated in the presence of the beads during 1 h at 4 °C for the immuno-depletion. The immuno-depleted extracts were then recovered after capture of the beads by a magnet. A control experiment using nude magnetic beads was also performed. The protein concentration within all extracts was assessed using the Pierce 660 nm Protein Assay Reagent (ThermoFisher Scientific) before the in vitro DNA ligation assay.

For the DNA ligation assay, we used the oligonucleotides LIG03 and Loop02 (Supplementary Table S4), to create a complex structure containing a single-strand break[75]. For the assay with protein extracts from *A. vaga*, the extracts were diluted in the dialysis buffer and used at the following final quantities in the assay: 0.64 µg, 0.32 µg, 0.16 µg, 0.08 µg, 0.04 µg, 0.02 µg, and 0.01 µg. A negative control corresponding to the dialysis buffer without protein was also added to the analysis. For the assay with protein extracts from HEK 293T cell lines, the extracts were also diluted in the dialysis buffer and used at the following final concentrations in the assay: 1.28 µg, 0.64 µg, 0.32 µg, 0.16 µg, 0.08 µg, 0.04 µg, and 0.02 µg. A negative control corresponding to the dialysis buffer without protein was also added to the analysis. Each ligation assay was performed in technical quadruplicate following the procedure described in Healing et al. [75] except that the reactions were performed for 2 h at 15 °C. Data were

presented with prism7 as mean ± SD of the four technical replicates. Data were background corrected by subtraction of the mean value obtained for the controls performed with the dialysis buffer. Background corrected absorbance values were normalized to the maximum absorbance value obtained during the assay. Curve fit was performed from non-linear regression analysis using the Sigmoidal, 4PL, X is log (concentration) fitting tool. DNA ligation activities were compared by doing a ratio of the slopes obtained in the linear portion of the activity curves for each reaction. Statistical analysis for comparison to the control conditions (non-irradiated or HEK SCR) was performed by Multiple unpaired t tests using the Holm-Sidak method, with alpha = 0.05 in prism7.

### Heterologous expression of AvLigE within the HEK 293 T human cell line

AvLigE-A and AvLigE-B genes were stably transfected within the HEK 293 T cell line by the CRISPR technology using the Stable cell line AVVS1 kit (Origene). HEK 293 T cell line was maintained at 37 °C and 5% $CO_2$ in Minimum Essential Medium Eagle (Sigma-Aldrich) supplemented with 10% of Fetal Bovine Serum and 1% of Penicillin/Streptomycin (ThermoFisher Scientific).

First, we inserted synthetic AvLigE-A and AvLigE-B genes (gblocks, IDT; Supplementary Table S4) within the pAAVS1-puro-DNR plasmid at the SgfI and MluI restriction sites using the Hifi DNA assembly cloning kit (New England Biolabs). This procedure cloned the genes of interest under the control of the constitutive CMV promoter. The whole cassette is bordered by two sequences homologous to the adeno-associated virus site 1 (AAVS1) locus on human chromosome 19, which is a safe insertion site within human cells. This safe harbor site is a transcriptionally active genomic region allowing robust and stable gene expression. This plasmid was co-transfected with the pCas-Guide vector following the instructions from Origene. Cas9 protein generates the DSB at the AVVS1 locus necessary to foster the insertion of the expression cassette at this genomic site. After several passages, the cell lines with stable insertions were selected by treatment with 0.5 µg/ml of puromycin. Several clones were selected and AvLigE-A/B insertions at AAVS1 locus were first verified by PCR using oligonucleotides amplifying the junction between the AAVS1 locus and the expression cassette (GE100033, Origene). Afterwards, we selected clones in which the mRNA expression of AvLigE-A and AvLigE-B was high and stable. In that purpose, total RNA from HEK 293T cell lines was first extracted using the Monarch total RNA Miniprep kit (New England Biolabs). The messenger RNAs corresponding to AvligE-A or AvLigE-B were then amplified and quantified by RT-qPCR using oligonucleotides described in Supplementary Table S4 and the Luna Universal qPCR Master Mix (New England Biolabs). The relative abundance of these transcripts was reported to the quantity of the human *gapdh* housekeeping gene (Supplementary Table S4). One clone expressing AvLigE-A or AvLigE-B which has passed these two quality control experiments was then selected for further experiments. A control cell line with an insertion of the empty pAAVS1-puro-DNR vector (HEK SCR) was also created.

To determine the proliferation potential of HEK 293T cell lines upon irradiation, we used the gold standard colony formation assay (CFA), which specifically test every cell in the population for its ability to undergo unlimited division[76]. For this assay, HEK 293T cell lines (SCR, AvLigE-A, and AvLigE-B) were seeded in 6-well plates 24 h before the irradiation experiments. X-ray radiations (0.002 and 0.004 kGy) were performed using a Precision X-ray X-RAD 225 XL instrument (225 kV, 13.5 mA, filter 1, 50 cm). After the irradiation, the cells were trypsinized and diluted at different concentrations to get isolated colonies. Counting of cells was performed using the Moxi Z Mini automated cell counter (Orflo). After 13 days of growth, the medium was removed above the cells, rinsed carefully with PBS, and incubated

30 min with 2 ml of a mixture of 6.0% glutaraldehyde and 0.5% crystal violet. The fixation and staining mixture was rinsed with tap water and the growing colonies were counted in each conditions. Small colonies (<50 cells) are formed but not scored for survival. Proliferation potential in each condition was measured by reporting the mean number of colonies growing at 0.002 and 0.004 kGy of X-rays to the mean number of colonies growing in the non-irradiated control. The experiment was performed in two biological replicates, each including three technical replicates ($n = 6$). Statistical significance between the control cell line (HEK SCR) and the cell lines expressing AvLigE-A/B under the variable irradiation conditions was determined by multiple $t$-tests using the Holm-Sidak method, with alpha = 0.05 using Prism 7 software.

## Reporting summary

Further information on research design is available in the Nature Portfolio Reporting Summary linked to this article.

## Data availability

Raw data are available in the Source Data file related to this manuscript. Protein sequences used in this study are available in the public repository NCBI NCBI, assembly ASM2161353v1. Mass spectrometry raw and processed data have been submitted to the PRIDE (Proteomics Identifications Database) repository with the dataset identifier PXD043051. Datasets used for phylogenetic analyses are available on Figshare: https://figshare.com/s/7cbed8c5d59abdd2f4ae. Data presented in this manuscript are protected under the patent WO 2023/161266 Methods and Compositions for increasing stress tolerance of cells and organisms (wipo.int). Source data are provided with this paper.

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

## Acknowledgements

The authors thank Catherine Demazy and Anne-Catherine Wéra for their generous help during the course of this study. The Morphim imaging platform and the Research Unit for Analysis by Nuclear Reactions (LARN) of UNamur are acknowledged for their technical help. This work was supported by the European Research Council (ERC) grant agreement 725998 (RHEA) to K.V.D., and from the Fédération Wallonie-Bruxelles via an'Action de Recherche Concertée' (ARC) grant to K.V.D. and B.H. E.N. obtained a FSR UNamur fund.

## Author contributions

E.N. conceived the experiments; E.N., P.S., M.G., M.C., J.V., M.L., A.B. and M.D. conducted the experiments; E.N., P.S., M.D., M.T., B.H., and K.V.D. analyzed the results. E.N., B.H., and K.V.D. wrote the manuscript. All authors reviewed the manuscript.

## Competing interests

The authors declare no competing interests.
