## [Peer Review File · Nature Communications]

Horizontal acquisition of a DNA ligase improves DNA damage tolerance in eukaryotesREVIEWER COMMENTS

Reviewer #1 (Remarks to the Author):

Please accept my apologies for the delay in returning my review. This is an interesting paper that applies a battery of strong lab techniques and phylogenetic analysis to describe a gene associated with the extreme radiation tolerance of bdelloid rotifers. The expression in human cells is a neat approach, partly circumventing problems in transforming rotifers at present, and partly bringing the hint of an applied angle (which I commend the authors for not over-selling in the manuscript). I think this is an interesting manuscript for this journal, but with one main caveat for further consideration.

The presence of HGT in bdelloids is well-established, and this gene certainly meets the criteria used in terms of grouping more closely with non-metazoan copies. But the phylogenetic analysis is not as clear cut as implied in the text. The tree has low bootstrap support across most nodes separating bdelloids from the nearest animals, which while not discounting HGT does require a bit more critical evaluation. Also, the topology of the tree in Figure 2 does not support the statements in the manuscript of ancestry of DNA ligase E in bacteria and parallel acquisition by many eukaryotes. The bacteria clades appear fairly derived in the wider tree. I have outlined points in detail below but, overall, a more critical evaluation of the phylogenetic evidence and alternative scenarios is required. Specifically, it would be good to check for phylogenetic conflict between different domains that might obscure a single strong signal for this gene.

I am less able to evaluate lab methods in detail, but they seem strong with adequate controls and interpretation is clear. The one other area to improve clarity is that results for copies A and B seem to differ a bit in the human expression than in the rotifer itself (i.e. which one seems most associated with radio-tolerance). It would be good to highlight and discuss this more. Other comments are listed below.

Line 20 – change “adaption” to “adaptation”.

Introduction

Line 23 to 25. Language is a bit imprecise and dramatic. “assaults” – is exposure to ionizing radiation an assault? “Aggression” is inaccurate, as this implies behavioural agency in the stress, which is lacking for ionizing radiation or chemical stresses. I suggest toning down these elements, so that even if they retain a lively style, they at least directly apply to the processes that you are talking about.

Line 37. Replace “being” with “are”. (to make this a complete sentence on its own)

Line 13. The introduction to horizontally acquired genes is not adequate to set the scene, given the importance of this feature in the present paper. Please expand to explain to an unfamiliar reader and also provide references to earlier papers, e.g. Gladyshev that first reported HGT with fairly comprehensive sequencing methods, Boschetti et al 2012 and Flot et al. 2013 that quantified the numbers of horizontally acquired genes in transcriptome and genome (not greatly changed by later studies), and Nowell et al. 2018 that confirmed the consistently high level across bdelloid species that differs from other related and distant comparable genomes. (All of these except the original Gladyshev paper are cited later on in the manuscript).

Results

Page 2. Line 37. This sounds interesting but I didn't understand what you meant about “visually affected”. Could this be associated with contraction of bdelloids into a tun upon exposure? Is that what you meant or something else. Please explain what you see visually.

Figure 1D. The 3rd and 5th panel shows the animal ‘upside down’, it would help to show all the pictures in the same orientation.

Line 44. "turned out to be an exogenous gene". Again, to say this is a headline part of your study and takes pride of place in the title, not much emphasis or detail here. What is the evidence? Actually I think this comes later, so maybe delete here but save for later section where the detail comes.

Line 50 – spelling "homoelogenous"

Page 3 Line 5 (figure 1D). There is an interesting pattern in the staining that the DNA ligase-B overlaps with nuclei in the soma but apparently not in middle of the body, which I think is the ovaries (which are stained with DAPI). This is apparent even at 24h. What might the significance of this be? Worth highlighting as not captured by the statement "an increased nuclear signal". (This does come up in the Discussion later, could flag in Results).

Line 12. More detail is needed to describe the results of the Alienomics pipeline, what scores does it give you that are the basis for concluding "strongly suggests" HGT?

Line 18 "Phylogenetic analyses revealed that members of this family are mainly found in bacteria (with 70 species being collapsed at the branch "Bacteria"), but other representatives were also identified in several non-metazoan eukaryote groups and in a few animals (Figure 2)." The word 'mainly' doesn't match figure 2, where around half of the sequences seem to be bacteria versus not. Equally, the topology of the tree does not convey that the eukaryote examples are simply nested in a wider diversity of bacterial sequences because the bacterial sequences occupy a derived position in the tree. From your tree, DNA ligase E does seem to be found quite widely among eukaryotes and even in several phyla of animals. The wording doesn't convey the result of the tree analysis accurately.

Line 24. This wording needs to be more critical as well. Absence in *Brachionus* could equally well represent loss from that lineage. It is worth pointing out its absence in its closest relatives, but that comparison alone is not enough to infer ancient HGT.

Line 30. There is quite a long discussion of DNA ligase K, as a putative related kind to DNA ligase E. It would be helpful to explain the taxonomic distribution of DNA ligase K to make complete sense of why you are discussing this ligase type. Is it bacterial and/or in eukaryotes or animals? It is shown as a collapsed clade on the tree but with no number of species shown. Did you include multiple sequences of this?

Line 36. "In addition to the core functional domains, AvLigE proteins contain additional poly(ADP-ribose)(PAR)-binding zinc fingers (PBZ) domains at the C-terminus which are not present in the bacterial proteins and other eukaryotic members of the family, suggesting a later acquisition during bdelloid rotifer evolution (Figure 2)" This confused me, because on Figure 2 there are PBZ domains in several other taxa, including 2 of them in a protist, and the other animal copies, albeit in a different order to the copies in the bdelloid versions. Please clarify. Do you mean not present in "some other eukaryote members" – I read it as meaning "all other eukaryotic members"?

Many nodes in the tree have bootstrap supports less than 50%, which is surprising given major differences in domain structure between different branches. This raised a few questions:

i) How strongly does the tree support polyphyly of bdelloid rotifers from the other animal copies? The only intervening node between the bdelloids and the other animals is a node with 91 bootstrap grouping the bdelloids with a protist. Some further test to reject monophyly might be useful given the lack of strong support for the deeper nodes in the tree.

ii) Given the domain structure of the whole gene, I wondered whether you tried running separate analyses for different domains. Perhaps they could show different signatures to each other. For example, a gene could contain a mixture of horizontally acquired domains and native animal domains. Do the blue or red domains in bdelloids have any similarity with the other animal blue and red domains, or with the blue domains in the DNA ligase 1,3,4,B,C,D in bdelloids? One possible explanation for the lack of strong phylogenetic signal is that different parts of the gene show conflicting phylogenetic signals.

It would be worth digging into these questions. At present the tree does not give the strong overwhelming signal for HGT that you might hope for (i.e. bdelloid copies nested well within non-animal copies and far away from nearest animal copies, with strong support for the whole phylogeny), or that some of the text implies.

The functional tests on AvLifE-B are very useful. For your inference about the A copy would have been neat to compare effects of that as well, but I imagine there were constraints for numbers of assays that were feasible etc.

Page 4. The story starts getting more complicated as until now focus was on copy B, but in the human expression it is copy A that seems to be expressed more and to convey radiation resistance. The wording in this section could help emphasize the contrast with earlier findings to guide the reader through. In the Discussion, you talk in general about differences in expression but don't highlight the contrast that B looks more important in rotifer, but A has largest effect in human cells.

Page 5. Line 19-21. Or that bdelloids acquired the gene secondarily from protists, fungi or chromista. Your tree would be more consistent with this hypothesis, otherwise parallel acquisition you would expect them each to be separately nested within your bacterial clade.

Line 36-39. Mmm, this sounds very speculative. Not sure a marine coastal habitat is especially stressful or more so than any other habitat occupied by all life on earth.

Page 7 line 5. "Gold models" doesn't mean anything – useful, interesting models..?

Page 9 line 10-12. Since exogenous acquisition is a major part of the story, more detail is needed on exactly how the method works to justify the inference here.

Line 24. Please provide the alignment length before and after exclusion of missing data. "Analyzed with Archaeopterix" – please say what analysis, or was this visualized rather than analyzed?

Figure 2 legend. Suggest deleting "manually" from the wording – it seemed odd when I read this, but I think you just mean these were added to the sequences separately from the initial blast search, so can just say you added sequences for the other families.

Reviewer #2 (Remarks to the Author):

The DNA damage response is fundamental for preserving genome integrity and the survival of organisms. Bdelloid rotifers are remarkable in that they recover from DNA damage caused by high doses of radiation. In this paper, the authors rely on a recent chromosome-scale assembly of the *Adineta vaga* genome to identify stress tolerance genes that might uncover the basis of radiation tolerance and DNA repair in these animals. Using comparative proteomics with irradiated *A. vaga*, they identify several upregulated proteins. This paper focuses on a DNA ligase homolog (which had the highest induction) which is also known to be induced by desiccation. There are four copies of this DNA ligase, and only peptides from copies B were detected.

Using their Alienomics pipeline, they strongly suggest that this IR-induced DNA ligase is a horizontally transferred gene. The phylogeny suggests that the protein belongs to the ligase E family of prokaryotic DNA ligases. Notably, the proteins have a PBZ domain that is absent in bacterial and many eukaryotic ligase E proteins, suggesting a later acquisition during bdelloid evolution. They also use mRNA sequencing to show that ligase E homologs are also upregulated following irradiation in *A. ricciae* and in the fungus *Mortierella verticillata*.

Perhaps the most significant result is determining whether AvLigE functions as a DNA ligase. Rather than knocking the AvLigE gene down or out, the authors did a ligation assay to show that protein extracts from irradiated *A. vaga* enhance DNA ligation activity, and more so when

compared to extracts with AvLigE depleted. Lastly, they demonstrate that when AvLigE is expressed in human cells, this protein improves radiation tolerance and affected morphology of irradiated cells.

While many studies have speculated about the mechanisms of DNA repair in bdelloids and possible proteins involved in the process, this study is significant because it provides the first evidence of a protein found in bdelloid rotifers (and not in other rotifers, such as monogononts) that plays an important role in DNA repair following exposure to irradiation. While genetic manipulation of bdelloids is challenging, the authors use a creative approach with in vitro ligation assays and expression studies using human cells to show that AvLigE is a functional DNA ligase. Evolutionary analysis also shed light on the potential origin of AvLigE by horizontal gene transfer, shedding light on the importance of this process in driving evolutionary change.

The paper is well-written and the approach and methods for this study are carefully planned and well thought out. The authors make use of a variety of analytical approaches (proteomics, qPCR, phylogenetics, in vitro ligation assays, etc.) to address the central objective of understanding the function of ligase E in bdelloids. The paper should be considered for publication, and below are comments regarding specific details in the paper that should be addressed before that point is reached.

Figure 1 legend: "Scheme A" should be "A."

Figure 1A: At t72h, "Gluthatione" should read "Glutathione"

Page 2, lines 25-28: The PFGE results are discussed first, followed by the proteomics (lines 28-33). However, in Figure 1 proteomics (1A) is shown before PFGE (1B). Consider reorganizing order of figures/text so that they better correspond.

Fig 1B: The PFGE shows striking evidence for DNA damage (at t0h) followed by DNA repair (at t4h, t24h and t72h). However, at t4h, t24h, and t72h, why does the repaired chromosomal DNA band not regain its original size (compared to the chromosomal band in the NI lane, which seems to be cutoff at the top of the gel, the dark bands in these lanes are smaller in size)? Does this imply that DNA repair is incomplete? Is there evidence that DNA repair is taking place accurately? For example, in this study do the animals recover from irradiation? Consider justifying this apparent discrepancy in the PFGE image when discussing Fig. 1B.

Page 2, Line 37: Is there a citation for the observation that muscle contraction is visually affected in irradiated bdelloids? Or is that an observation from this paper? This is a very interesting detail, but it is unclear where this has been observed.

Page 2, Line 38: Earlier on page 2 (lines 32-33), the authors note that PNKP, DNA PolB and PARP proteins are upregulated following irradiation. Later (line 38), they state that all these proteins are upregulated at t24h and t72h. However, PARP3 is not included in the proteins that remain upregulated at t24h and t72h post-irradiation.

Fig. 1D: It is surprising that DNA ligase-B is barely detectable in the t4h individual, especially since proteomic and western blot results suggest there should likely be more protein present. The authors note that immunofluorescence shows an increased nuclear signal at t24h which then decreases at t48h and t72h. However, the level of DNA ligase-B nuclear localization seems underwhelming, considering the total amount of DNA ligase-B signal at 24, 28 and 72h that does not localize to the nuclei. The authors should address the significance of the DNA ligase-B that is not localized to the nucleus. Also, DAPI (nuclei) staining is much brighter at t24h compared to other time points (e.g. non-irradiated)? Could some cellular process occurring during recovery from irradiation that would lead to this outcome?

Figure 2: This is such an unusual grouping of taxa in the ligase E lineage. The authors do an excellent job addressing this with *Mortierella verticillata* and provide a novel analysis of radiation tolerance in this species and demonstrate that MvLigE is also upregulated following irradiation. However, is anything known about the DNA repair capacity of *Rotaria* sp.? Two *Rotaria* species

have ligase E-B homologs but are noted as being desiccation-sensitive. A statement about why these two species would maintain ligase E would be helpful.

Page 4 (lines 11-12): The authors state "To determine whether this horizontally acquired ligase may have a similar adaptive role as the one proposed for AvLigE...." Are the authors suggesting that MvLigE was also horizontally transferred in an independent event? If so, is there evidence from a similar Alienomics pipeline that was used for *A. vaga*? Or this is based solely on the phylogenetic tree? If so, evidence from the tree alone needs to be better explained. The long branches (which could yield long branch attraction artifacts) and lack of bootstrap support for relationships within the ligase E lineage makes the claim of horizontal gene transfer difficult to understand. A similar issue is in the abstract (page 1, lines 16-17) where the authors state (...and its horizontal acquisition by bdelloid rotifers and other eukaryotes that also exhibit radiation tolerance...). They do not appear to provide evidence for horizontal transfer of ligase E in other eukaryotes (aside from the phylogenetic tree) and clarification needs to be addressed here.

Page 5 (lines 35-39): The authors mention other animals that have ligase E and note that they live in "marine coastal or shallow-water ecosystems that are stressful environments..." Clarification about what makes these environments stressful (in reference to DNA damage and the benefit of harboring a ligase E gene) would be helpful.

Fig. 3D. Unclear what this shows (the image with 6 panels and different levels of IR). Needs to be clearly explained in figure legend.

Fig. 4D: Is there a significant difference between relative absorption when comparing the blue and green data points at each protein extract concentration? Some statistical analysis (as done for Fig. 5C) should be included to better demonstrate that the reduction in absorbance was significantly lower for the AvLigE-B depleted sample (compared to the IR sample) for corresponding extract concentrations.

Fig. 5B. Same comment as for Fig. 4d (some sort of statistical analysis should be included).

Fig. 5C: Could the increase in survival for cells expressing AvLigE be the result of overexpression of ligase? For example, if the authors included a cell line expressing ligase 1 from *A. vaga* (or even a ligase from humans) and observed a similar result, this could suggest that overexpression of DNA ligase, in general, improves survival from irradiation. However, if they did that and only AvLigE showed improved survival (as in Fig. 5E) then this would be more convincing evidence that AvLigE (and not simply ligase overexpression) specifically improves radiation tolerance. It is recommended that the authors include a trial with a ligase other than AvLigE to demonstrate the improved radiation tolerance is not an artifact of ligase overexpression.

Page 6, Line 7: "antioxydant" should read "antioxidant"

Page 6, lines 37-50: The authors provide a compelling description for a mechanism of DNA repair in bdelloids. However, much of this speculation is based on unpublished findings from Terwayne et al. (via personal communication). For example, this section refers to somatic nuclei providing the strongest signal in the proteomic analysis (but this is not referred to in the manuscript) and a time window of DNA repair during oogenesis in which a meiosis-like configuration is present and different repair strategies for somatic and germline cells. These are intriguing findings but data to support these claims are not presented, not published elsewhere. Why were some of these data not included in this study to make for a more complete story of DNA repair? The authors should consider placing less emphasis on unpublished results at the end of their discussion section, or provide more data to support these statements.

Page 9, line 15: Three bdelloid genome assemblies are mentioned. However, the authors also note that ligase E homologs are absent in the monogonont *Brachionus plicatilis* genome. Is there an assembly for this?

Page 9, line 35: Was qPCR for bdelloids done in triplicate (as for *M. verticillata* mentioned below; line 47)?

Reviewer #3 (Remarks to the Author):

The authors compare the proteome of the bdelloid rotifer *Adineta vaga* exposed to 1Gy of X irradiation with unexposed animals and find that a small number of proteins increase in abundance by > 2.5 fold. Among these is a protein encoded by a gene that is of non-Metazoan origin—one of the many “alien genes” that have been adopted by bdelloids. The authors characterize this gene as “ligase E–B,” one of a pair of genes differing in the number of PBZ domains (1 in “A” and 2 in “B”), with only the B protein being over-represented after irradiation. The authors support this result with qPCR of the two gene copies and of other ligases. They also show that a potential homolog in a fungus, with structural similarity to the A copy, is overexpressed after exposure to X-irradiation. The authors go on to use an antibody to the B protein to examine the localization of the protein in whole animals and to deplete cell lysates of the protein. They use a single-strand break assay to show that depleted lysates from irradiated animals have a decreased rate of repair compared to undepleted lysates. Finally, they transfect human tissue culture cells with the A and B copies and demonstrate that each increases the resilience of the cells to X-irradiation.

The application of proteomics to understand bdelloid radiation tolerance is a new and important advance, with significance beyond this particular ligase. The validation that this gene encodes a functioning ligase with a role in resilience to extreme DNA damage is the first proof of the hypothesis that this gene and other alien DNA repair genes acquired by bdelloids confer additional DNA repair capability. The demonstration that the ligase functions in human tissue culture cells is significant in its own right, and also opens up a new avenue for studying not only this ligase but the other proteins identified as playing a role in the remarkable radiation resilience of bdelloid rotifers.

The gene was previously identified as a horizontally transferred ligase with potential to enhance bdelloid DNA repair in 2018 by Hecox-Lea and Mark-Welch (ref 27), who called the gene “LigK” with the same designation of A and B for the copies with one and two PBZ domains, respectively. They also presented a phylogenetic analysis demonstrating horizontal transfer into *A. vaga*, with a similar summary of domain structure in representative species, as well as RNA-Seq results showing that the transcript abundance of both copies increases in response to desiccation, with the B copy increasing about twice as much as the A copy. These points could be better leveraged in the manuscript.

The statement in the abstract that “ligase E is by far the major contributor of DNA breaks ligation activity” is not supported by the evidence in the paper. Depletion of the B copy resulted in a reduction of ~35% in the rate of repair in a specific single strand break assay. While unquestionably significant, there is plenty of room for other major players even in this specific role, and of course there are many additional roles for ligases in DNA repair.

Throughout the manuscript proteins are described as “up regulated” or “over expressed” or “differentially expressed” when proteomics or western blot shows an increase of protein levels. Given the numerous phenomena that could result in overabundance (change in transcript rate, mRNA half-life, translation efficiency, protein half-life, etc) it would be more prudent to use a term like “over-represented” as the authors do in “Another protein, the alpha-protein kinase *vwka*, was also over-represented in the irradiated samples” or “induction” as used in the next paragraph.

The authors use a polyclonal antibody to the B protein. I do not see any demonstration of its specificity. Does it bind to the A protein? The A protein is ~51kD, a difference that can be resolved on the sort of gel shown in Fig4B. Can the gel image in Fig 4B be expanded to demonstrate whether the antibody binds to A? If it does bind to A I don't think this changes the overall significance of the results, the wording would just need to be tweaked to make it less specific.

The authors find that human tissue culture cells transfected with copy A have greater repair activity than those with copy B, and in the Discussion speculate why the A protein would be more active than B in human cells. This seems to be based on the assumption that the A protein is less active than B in *A. vaga*. But this may not be the case; the assays with *A. vaga* only involved depletion of B. The A protein could be more active than B in *A. vaga*. There is simply

more of B produced in response to irradiation. Perhaps because it isn't as efficient as A.

Fig 1A: "Protein with PARP domain" is listed only at 4 hours. In Fig S1 it is listed at 24 hours after 800Gy (Fig S1). Is it in the other timepoints below the arbitrary 2.5x threshold? Similarly PARP2 is present at 4 and 24 hours but not 72 hours, and not in Fig S1. Perhaps Table S1 could be mentioned in the legend or the results section and include more data? This would help solidify the contribution as the first proteomic investigation of bdelloid radiation exposure.

Fig 1B: This is not an analysis, just a gel image. The legend should read something like "PFGE image showing DNA DSB repair following irradiation." Also, I'll point out here that this image and the general discussion of irradiation-produced DSBs is somewhat at odds with the data, which show that the ligase is capable of repairing single strand nicks. Of course there are presumably plenty of these nicks produced by 1kGy of X-rays, but the connection between this ligase and DSB repair is circumstantial.

Fig 1D: I don't know what to make of this figure. It does show the presence of the protein *in situ* over a time course consistent with Figs 1A and C, and presumably shows that the B protein is confined to nuclei. But in the overlay images I'm not seeing anything like a 1:1 correspondence between DAPI and immunofluorescence. There seem to be red spots without corresponding green spots and vice versa. Green without red could be simply brighter green signal, but it looks like there are entire organs without B protein. Could expression of copy B be confined to certain cells? That would certainly be interesting! Also, only one image of each time point is shown so it is impossible to evaluate the assertion that signal "appears to decrease at t48h and t72h." I would point out that this interpretation is at odds with the increase in the abundance of copy B shown in Figs 1A and C.

Fig 2: I am not convinced that the "Ligase E" clade is meaningful. This is an unrooted tree and there is no clade containing the eponymous bacterial Ligase E and the other sequences labeled as "Ligase E"; the only support is that they are all not in the other ligase clades. There is as much sequence diversity in the proposed clade as there is between all other ligase clades combined, and unlike other ligase families there are drastically different domain architectures within the clade, implying very different properties and activities (the canonical Ligase E gene from bacteria contains only the two domains colored blue and green in the figure, and a signal peptide for export, and is associated primarily with repair of nicks). So in addition to not being a true clade it is unlikely to be a useful distinction that will help us designate or understand ligases. I agree that the original name of "Ligase K" is also not ideal; it was chosen because all the genes designated LigK in ref 27 had top hits to conserved domains Adenylation_kDNA_ligase_like (cd07896) and OBF_kDNA_ligase_like (cd08041) in the NCBI CDD database available at the time, but this isn't true of newer database releases.

That the ligase genes are alien was already demonstrated in ref 27 (albeit with fewer sequences) so the figure is not necessary for that purpose. The figure does include three additional bdelloid species, which helps identify the horizontal transfer as a single event early in bdelloid evolution, an important point the authors note. It also includes the protozoan *P. fungivorum*, which is the closest known sequence and the only non-bdelloid example of the PBZ domain architecture seen in the B copy (though the additional peroxidase domain is certainly unusual). When I look at the gene tree I see phyletic incongruence, patchy distribution, and low resolution, implying that the core blue/green domains have been passed around and adapted for different purposes by different lineages, which brings into question the whole concept of a meaningfully "homologous" gene family.

Fig 3A: In the context of this panel the authors may want to mention that desiccation also increases transcript abundance of copies A and B, with B increasing twice as much as A, but that the magnitude of the change in Ligase B is *much* greater after irradiation. This connects the irradiation work done here with the more biologically relevant process of desiccation.

Methods:

"were lyzed by 1,000 Dounce with tight pestle" missing a word?

"proteic" isn't a common word in English. Maybe "peptide"?

Reviewer #4 (Remarks to the Author):

In this study Nicolas and colleagues characterized a prokaryotic DNA ligase involved in DNA damage tolerance. Mass spectrometry proteomic analysis of a rotifer organism irradiated for different times points led to the identification of a DNA ligase as the most, according to the authors, upregulated protein post-irradiation. Other proteins with potential roles in regulating DDR were also identified in the screen. The analysis was then focused on characterising DNA ligase expression in related species and in testing the hypothesis that this enzyme confers tolerance to DNA damage by heterologous expression in human cells.

Major issues

1. The analysis and dissemination of proteomics data does not conform to standards in the field. The mass spectrometry raw data has not been submitted to a repository (e.g. PRIDE) and processed data for all identifications (identities, sequences and LF quantity values for all identified peptides) should be provided in supplementary datasets. Without these data, it is not possible to assess the quality of data that is reported in the article.
2. There are no information on the database that was used for protein identification from mass spectrometry data. How many sequences does it contain and how were the sequences translated?
3. The proteomic experiment seems to have been carried out once without technical or biological replicates.
4. There is no justification for the threshold for considering differentially regulated proteins to be set to 2.5 fold.
5. There are no details of how the normalization of the LFQ data was carried out.
6. The Supplemental tables are not informative. For example, data in Table S1 is not informative because the acronyms in the headings are not defined in a table heading or caption.

REVIEWER COMMENTS

Reviewer #1 (Remarks to the Author):

Please accept my apologies for the delay in returning my review. This is an interesting paper that applies a battery of strong lab techniques and phylogenetic analysis to describe a gene associated with the extreme radiation tolerance of bdelloid rotifers. The expression in human cells is a neat approach, partly circumventing problems in transforming rotifers at present, and partly bringing the hint of an applied angle (which I commend the authors for not over-selling in the manuscript). I think this is an interesting manuscript for this journal, but with one main caveat for further consideration.

The presence of HGT in bdelloids is well-established, and this gene certainly meets the criteria used in terms of grouping more closely with non-metazoan copies. But the phylogenetic analysis is not as clear cut as implied in the text. The tree has low bootstrap support across most nodes separating bdelloids from the nearest animals, which while not discounting HGT does require a bit more critical evaluation. Also, the topology of the tree in Figure 2 does not support the statements in the manuscript of ancestry of DNA ligase E in bacteria and parallel acquisition by many eukaryotes. The bacteria clades appear fairly derived in the wider tree. I have outlined points in detail below but, overall, a more critical evaluation of the phylogenetic evidence and alternative scenarios is required. Specifically, it would be good to check for phylogenetic conflict between different domains that might obscure a single strong signal for this gene.

I am less able to evaluate lab methods in detail, but they seem strong with adequate controls and interpretation is clear. The one other area to improve clarity is that results for copies A and B seem to differ a bit in the human expression than in the rotifer itself (i.e. which one seems most associated with radio-tolerance). It would be good to highlight and discuss this more. Other comments are listed below.

We thank Reviewer #1 for his very positive assessment about our work and constructive comments on the manuscript. Answers to his/her concerns regarding the phylogenic analysis of LigE proteins and the respective contribution of AvLigE-A and B on radiotolerance are detailed below.

Line 20 – change “adaption” to “adaptation”.

Done (Page 1, Line 21)

Introduction

Line 23 to 25. Language is a bit imprecise and dramatic. “assaults” – is exposure to ionizing radiation an assault? “Aggression” is inaccurate, as this implies behavioural agency in the stress, which is lacking for ionizing radiation or chemical stresses. I suggest toning down these elements, so that even if they retain a lively style, they at least directly apply to the processes that you are talking about.

The text in the introduction has been changed by (Page 1, Line 24): “All living organisms endure constant endogenous and exogenous stresses. Under a stressful environment, the survival potential of an individual depends on its capacity of protecting its cellular components from those threats and on its ability to repair the incurred damages”.

Line 37. Replace “being” with “are”. (to make this a complete sentence on its own)

Done (Page 1, Line 38).

Line 13. The introduction to horizontally acquired genes is not adequate to set the scene, given the importance of this feature in the present paper. Please expand to explain to an unfamiliar reader and also provide references to earlier papers, e.g. Gladyshev that first reported HGT with fairly comprehensive sequencing methods, Boschetti et al 2012 and Flot et al. 2013 that quantified the numbers of horizontally acquired genes in transcriptome and genome (not greatly changed by later studies), and Nowell et al. 2018 that confirmed the consistently high level across bdelloid species that differs from other related and distant comparable genomes. (All of these except the original Gladyshev paper are cited later on in the manuscript).

In the introduction we have added the following sentence and references, as suggested by reviewer 1 (Page 2, Lines 13-15): “Indeed, previous studies have demonstrated that bdelloid rotifers harbor a large amount of foreign genes coming from bacteria, fungi and plants, often clustered in subtelomeric regions along with transposable elements and reaching > 6% across studied bdelloid genomes (Gladyshev et al., 2008; Boschetti et al., 2012; Flot et al. 2013; Nowell et al. 2018)”.

Results

Page 2. Line 37. This sounds interesting but I didn’t understand what you meant about “visually affected”. Could this be associated with contraction of bdelloids into a tun upon exposure? Is that what you meant or something else. Please explain what you see visually.

Most of the irradiated rotifers do not turn into tun shape as seen during desiccation. Rather, after irradiation, the individuals present a spasmodic-like contraction phenotype, suggesting muscle contraction defects. This precision has been added on Page 2, lines 40-43 of the manuscript.

Figure 1D. The 3rd and 5th panel shows the animal ‘upside down’, it would help to show all the pictures in the same orientation.

Done.

Line 44. “turned out to be an exogenous gene”. Again, to say this is a headline part of your study and takes pride of place in the title, not much emphasis or detail here. What is the evidence? Actually I think this comes later, so maybe delete here but save for later section where the detail comes.

In the new sentence we do not mention the exogenous gene, it has therefore been changed to (Page 2, Line 49; Page 3, Line 1): This ligase was also shown by transcriptomic analysis³⁰ to be strongly induced upon desiccation treatment and was characterized into further details here.

Line 50 – spelling “homoelogenous”

Done (Page 3, Line 5).

Page 3 Line 5 (figure 1D). There is an interesting pattern in the staining that the DNA ligase-B overlaps with nuclei in the soma but apparently not in the middle of the body, which I think is the ovaries (which are stained with DAPI). This is apparent even at 24h. What might the significance of this be? Worth highlighting as not captured by the statement “an increased nuclear signal”. (This does come up in the Discussion later, could flag in Results).

This is an important observation that is indeed worth pointing out in the results. It is in accordance with the observations made by Terwagne *et al.* (PMID: 36449626) showing that the oocytes are not stained with the nucleotide analog F-ara-Edu (used here as a readout for *de novo* DNA synthesis associated to DNA repair) during or after exposure to ionizing radiation. Instead, DNA repair in the oocytes is delayed to a specific time window of oocyte maturation during which homologous chromosomes adopt a meiotic-like juxtaposed configuration. In the somatic cells however, Terwagne *et al.* observed that DNA repair occurred immediately after IR exposure with all nuclei being stained with F-ara-Edu. Repair of the DNA DSBs in the germ line of *A. vaga* appear to use the meiotic homologous recombination machinery to promote complete genome reassembly, while in the somatic nuclei the DSBs are most likely to be sealed by nonhomologous end joining (NHEJ) that generally predominates in non-dividing cells. Since the Ligase E proteins (and other ligases) are typically involved in the ligation steps of DNA repair pathways, and since the oocytes in *A. vaga* are not getting repaired immediately after exposure to IR, we do not expect to observe Ligase E labeling in the oocytes of *A. vaga*. These specific germline nuclei are located in both ovaries in the middle section of the adult body, and this may explain why we did not observe DNA-ligase B labeling in this part of the body.

The position of the ovaries is now highlighted in Figure 1C by a dashed line. In the result section, we specify this absence of labeling of the ovaries by DNA ligase E copy B. We added the following sentences (Page 3, Line 15-18): “However, virtually no DNA ligase B labeling was observed in the middle part of the adult body where the ovaries and germline are localized (circled by dashed lines in Figure 1C and Figure S1D). The IR-induced expression pattern of this DNA ligase B and its nuclear localization in *A. vaga*, strongly suggests that it might be a key actor involved in bdelloid rotifer DNA repair in somatic nuclei.” In the discussion (Page 7, Line 23) we make the link with the results of Terwagne *et al.*, 2022 (PMID: 36449626). In the new Figure S1D we also show an enlargement on somatic and germline cells to show the difference in AvLigE copy B localization between both cell types.

Line 12. More detail is needed to describe the results of the Alienomics pipeline, what scores does it give you that are the basis for concluding “strongly suggests” HGT?

In order to investigate whether *Adineta vaga* ligases E stemmed from a potential HGT event (or not),

we used our 'Alienomics' tool (to be published elsewhere, see also PMID: 34613768). Briefly, Alienomics measures various characteristics for every gene in a given genome and the resulting scores are then taken into account to classify each gene into different categories, including the “expressed HGT” category. The table below shows the Alienomics results for the two copies of ligase E in *A. vaga*. This table has also been added to the manuscript as Table S2.

gene_ID	scaffold	category	start	end	BLAST score	GC score (%GC)	coverage score	expression score (TPM)
UJR14855.1	Chrom_1	expressed HGT	5793041	5794788	-0.977	0 (35.1%)	0	1 (2.4)
UJR31854.1	Chrom_4	expressed HGT	7344188	7345745	-0.999	0 (34.7%)	0	1 (3.5)

The categorization of the two ligase genes is straightforward: BLAST scores around -1 (the smallest possible score) represent very strong best-hit on non-metazoan sequences; coverage score at 0 indicates that read coverage is the same as the average read coverage observed along the genome (indicating that the gene is integrated within *A. vaga* chromosomes); expression score of 1 indicates that this gene is expressed in hydrated condition (based on RNA-Seq data) though at a very low level (2.4 and 3.5 transcripts per million, respectively). Finally, a GC score of 0 indicates that the GC content of these Ligase E genes is not sufficiently different (35%) from the GC content observed in the rest of the genome (31%) to solely suggest an HGT origin. This latter result suggests that these HGT candidates are probably ancient within bdelloid rotifers: GC content likely had time to be homogenized with the GC content of its host genome.

We have added in the manuscript on Page 3, Lines 22-23 the main results of this Table S2.

Line 18 “Phylogenetic analyses revealed that members of this family are mainly found in bacteria (with 70 species being collapsed at the branch "Bacteria"), but other representatives were also identified in several non-metazoan eukaryote groups and in a few animals (Figure 2).” The word ‘mainly’ doesn’t match figure 2, where around half of the sequences seem to be bacteria versus not.

The phylogenetic tree of ligase E (Figure 2 in the manuscript) includes: 12 metazoa (including 8 bdelloid rotifers), 10 protozoa, 12 chromista, 43 fungi and 78 bacteria.

We agree with the reviewer that the wording “mainly” might seem a bit “off” when looking at Figure 2, although bacterial sequences do represent the majority of ligase E sequences. For clarity, Figure 2 presents a reduced taxonomic sampling of 155 sequences out of the 502 hits retrieved from an initial protein blast against uniref50 database (the threshold set at 155 overall best hits was selected arbitrarily, see Methods section).

A phylogenetic tree including all 502 ligase sequences was added to the manuscript as Figure S2A. Among these 502 sequences, ≥ 69% are from bacteria.

We added the following information on Page 3, Lines 30-31: “Phylogenetic analyses revealed that members of this family are mainly found in bacteria (> 50% of species collapsed at the branch “Bacteria” in Figure 2 and >69% are bacteria in Figure S2)”.

Equally, the topology of the tree does not convey that the eukaryote examples are simply nested in a wider diversity of bacterial sequences because the bacterial sequences occupy a derived position in the tree. From your tree, DNA ligase E does seem to be found quite widely among eukaryotes and even in several phyla of animals. The wording doesn’t convey the result of the tree analysis

accurately.

We expect that the complete phylogenetic tree now provided in Figure S2A (see our response above) will convince the reviewer that eukaryotic ligase E sequences appear nested within many paraphyletic bacterial lineages. We agree with the reviewer that the exact taxonomic origin of the AvLigE copies is uncertain given the topology (and bootstraps) of the phylogenetic trees: these genes could have been acquired from bacteria, fungi or other eukaryotes. We thus refrain from any interpretation regarding the taxonomic origin of the HGT event within bdelloid rotifers. We have now mentioned in the result section (Page 3, Lines 38-39): “The exact origin of this Ligase E gene in bdelloid rotifers is uncertain, its closest relative appears to be *Planoprotostellium* in all phylogenies obtained (Figures 2 and S2)”. However, it seems more likely that the evolutionary origin of the entire ligase E lineage is bacterial.

Line 24. This wording needs to be more critical as well. Absence in *Brachionus* could equally well represent loss from that lineage. It is worth pointing out its absence in its closest relatives, but that comparison alone is not enough to infer ancient HGT.

We infer that the Ligase E genes in bdelloid rotifers stemmed from an ancient HGT because they are found in all bdelloid species whose genome was sequenced, belonging to different bdelloid rotifer genera and families, suggesting that this Ligase E gene was acquired before the split of the bdelloid families. If the monogonont rotifer *Brachionus plicatilis* had a ligase E and then lost it (or if we missed it during either sequencing, assembly, gene prediction or homology search) its origin would still suggest a HGT because its closest relative appears to be *Planoprotostellium*. In fact, if *Brachionus* had a ligase E and then lost it, we would have inferred an even more ancient origin for this HGT event (not only in the ancestor of bdelloid rotifers, but possibly in the ancestor of all rotifers).

Line 30. There is quite a long discussion of DNA ligase K, as a putative related kind to DNA ligase E. It would be helpful to explain the taxonomic distribution of DNA ligase K to make complete sense of why you are discussing this ligase type. Is it bacterial and/or in eukaryotes or animals? It is shown as a collapsed clade on the tree but with no number of species shown. Did you include multiple sequences of this?

We thank the reviewer because this section indeed requires more explanation. DNA ligases K are exclusively found in kinetoplastids while DNA ligase E was not detected in kinetoplastids. The lack of taxonomic overlap between ligases K and E is a pivotal argument supporting the hypothesis that DNA ligase K might actually represent the DNA ligase E of kinetoplastids. Since the DNA ligase K of kinetoplastids evolved quickly, they are quite divergent from other ligase E (or K) and likely suffer from both a lack of phylogenetic signal and a potential long branch attraction artifact during phylogenetic inference. Note that in the complete phylogenetic tree of Figure S2A, DNA ligase K of kinetoplastids are placed within a ligase E lineage, suggesting that all these ligases might have the same evolutionary origin (i.e., within bacteria) and that researchers might want to give them the same name. We rephrased the entire paragraph in the result section (Page 3, Lines 43-51).

We added ligase K sequences to our phylogeny because in the manuscript by Hecox-Lea and Mark Welch, 2018, the ligase E sequences of *Adineta vaga* were annotated as Ligase K (see Result section, Page 3, Line 43). As stated in Table S3, the sequences of 4 different ligases K were used in this analysis. None of them were retrieved from the blastp analysis, likely because they are too divergent from the AvLigE-B sequence. To infer the phylogenetic relationship between Ligase E and Ligase K, we randomly choose 4 sequences of described ligases K from kinetoplastids.

Line 36. “In addition to the core functional domains, AvLigE proteins contain additional poly(ADP-

ribose)(PAR)-binding zinc fingers (PBZ) domains at the C-terminus which are not present in the bacterial proteins and other eukaryotic members of the family, suggesting a later acquisition during bdelloid rotifer evolution (Figure 2)” This confused me, because on Figure 2 there are PBZ domains in several other taxa, including 2 of them in a protist, and the other animal copies, albeit in a different order to the copies in the bdelloid versions. Please clarify. Do you mean not present in “some other eukaryote members” – I read it as meaning “all other eukaryotic members”?

The reviewer is correct. We have clarified this in the result section (Page 4, Lines 4-7) mentioning: "In addition to the core functional domains, AvLigE proteins contain additional poly(ADP-ribose)(PAR)-binding zinc fingers (PBZ) domains at the C-terminus which are not present in the bacterial and in most other eukaryotic proteins (except *Thecamonas trahens* and *Mortirella verticillata*), although three other animal species do contain PBZ domains at the N-terminus of their Ligase E, suggesting a later acquisition during metazoan evolution (Figure 2).

Many nodes in the tree have bootstrap support less than 50%, which is surprising given major differences in domain structure between different branches. This raised a few questions:

- i) How strongly does the tree support polyphyly of bdelloid rotifers from the other animal copies? The only intervening node between the bdelloids and the other animals is a node with 91 bootstrap grouping the bdelloids with a protist. Some further test to reject monophyly might be useful given the lack of strong support for the deeper nodes in the tree.
- ii) Given the domain structure of the whole gene, I wondered whether you tried running separate analyses for different domains. Perhaps they could show different signatures to each other. For example, a gene could contain a mixture of horizontally acquired domains and native animal domains. Do the blue or red domains in bdelloids have any similarity with the other animal blue and red domains, or with the blue domains in the DNA ligase 1,3,4,B,C,D in bdelloids? One possible explanation for the lack of strong phylogenetic signal is that different parts of the gene show conflicting phylogenetic signals.

We thank Reviewer #1 for this important comment. We have performed new phylogenetic analyses for the different protein domains: ATP; OB; ATP + OB and we included supplemental Figure S2B presenting the phylogenetic tree with the ATP+OB fold domains alone. A sentence describing these new results is given Page 3, Line 51; Page 4, Lines 1-3.

Several phylogenetic analyses were run using various subparts of the ligase alignment but also using various evolution models. We observed no strong phylogenetic conflict between any pair of analyses and therefore do not suspect any conflicting phylogenetic histories between protein domains. Indeed, in all 5 topologies (OB, ATP, ATP+OB, full Ligase gene including 155 and 502 sequences) the phylogenetic signal for deep nodes within Ligases E is low, except for a few relationships including the sister-clade relationship between *Planoprotostellium* and bdelloid rotifers. As a consequence, the polyphyly of metazoans is strongly supported in all analyses since bdelloid rotifers are always sister-clade to the fungus *Planoprotostellium fungivorum* (with a high bootstrap support) and all bdelloid rotifer species always form a strongly supported monophyletic clade, never grouping with the other animal species.

It would be worth digging into these questions. At present the tree does not give the strong overwhelming signal for HGT that you might hope for (i.e. bdelloid copies nested well within non-animal copies and far away from nearest animal copies, with strong support for the whole phylogeny), or that some of the text implies.

We agree we cannot infer the taxonomic origin of the HGT event within bdelloid rotifers but since bdelloid copies nested well within non-animal copies (Cfr the different trees mentioned above) and since the Alienomics analysis strongly suggests that Ligase E is a HGT gene (see table above) within bdelloid rotifers, we still propose that Ligase E was likely acquired horizontally by the ancestor of bdelloid rotifers.

The functional tests on AvLigE-B are very useful. For your inference about the A copy would have been neat to compare effects of that as well, but I imagine there were constraints for numbers of assays that were feasible etc.

Since copy A of AvLigE is not expressed in bdelloids (not retrieved in our proteomic or Western blot analysis – Figure 1, and weakly detected in the qPCR analysis - Figure 3), it was not possible to deplete AvLigE copy A protein or perform any functional assay on it.

Page 4. The story starts getting more complicated as until now focus was on copy B, but in the human expression it is copy A that seems to be expressed more and to convey radiation resistance. The wording in this section could help emphasize the contrast with earlier findings to guide the reader through. In the Discussion, you talk in general about differences in expression but don't highlight the contrast that B looks more important in rotifer, but A has the largest effect in human cells.

In the discussion (Page 6, Lines 48-50; Page 7, Lines 1-7) we emphasize on the fact that copy A of AvLigE has a larger effect on the survival of human cells when compared to copy B, while we only detected an increased expression of copy B in irradiated *A. vaga* individuals. We also mentioned that the full explanation of the difference between the ligation activity of AvLigE-A and -B in human cells vs bdelloid rotifers requires further investigation.

Page 5. Line 19-21. Or that bdelloids acquired the gene secondarily from protists, fungi or chromista. Your tree would be more consistent with this hypothesis, otherwise parallel acquisition you would expect them each to be separately nested within your bacterial clade.

We agree with Reviewer #1 that we should be more careful on the origin of horizontal gene transfer for Ligase E in bdelloid rotifers. The result section on the phylogenetic analysis was re-written in the revised manuscript, taking into account the new phylogenetic analyses on distinct protein domains that have confirmed *Planoprotostellium* Ligase E as sister clade to the Ligases E of bdelloid rotifers (see above), while the exact origin of this Ligase E in bdelloid rotifers remains uncertain.

Line 36-39. Mmm, this sounds very speculative. Not sure a marine coastal habitat is especially stressful or more so than any other habitat occupied by all life on earth.

We have added selected references here that support the higher oxidative stress encountered in marine coastal habitats (Page 6, Lines 16-19), but we agree it remains very speculative and discuss it as such.

It is also important to note that the 4 Ligase E sequences belonging to other metazoans show problematic relationships since they do not reflect the known species phylogeny among metazoans (e.g. paraphyletic protostomes).

Page 7 line 5. “Gold models” doesn’t mean anything – useful, interesting models..?

We replaced it by: interesting model systems (Page 7, Line 35)

Page 9 line 10-12. Since exogenous acquisition is a major part of the story, more detail is needed on exactly how the method works to justify the inference here.

More details on the approach and the results are given in a previous answer (see above) and in Table S2

Line 24. Please provide the alignment length before and after exclusion of missing data. “Analyzed with Archaeopterix” – please say what analysis, or was this visualized rather than analyzed?

Alignment length information are now indicated in the main text (Page 10, Lines 17-19). We clarified that we indeed used Archaeopterix simply to visualize the phylogenetic tree, which is not an "analysis" (Page 10, Line 20).

Figure 2 legend. Suggest deleting “manually” from the wording – it seemed odd when I read this, but I think you just mean these were added to the sequences separately from the initial blast search, so can just say you added sequences for the other families.

Done.

Reviewer #2 (Remarks to the Author):

The DNA damage response is fundamental for preserving genome integrity and the survival of organisms. Bdelloid rotifers are remarkable in that they recover from DNA damage caused by high doses of radiation. In this paper, the authors rely on a recent chromosome-scale assembly of the *Adineta vaga* genome to identify stress tolerance genes that might uncover the basis of radiation tolerance and DNA repair in these animals. Using comparative proteomics with irradiated *A. vaga*, they identify several upregulated proteins. This paper focuses on a DNA ligase homolog (which had the highest induction) which is also known to be induced by desiccation. There are four copies of this DNA ligase, and only peptides from copies B were detected.

Using their Alienomics pipeline, they strongly suggest that this IR-induced DNA ligase is a horizontally transferred gene. The phylogeny suggests that the protein belongs to the ligase E family of prokaryotic DNA ligases. Notably, the proteins have a PBZ domain that is absent in bacterial and many eukaryotic ligase E proteins, suggesting a later acquisition during bdelloid evolution. They also use mRNA sequencing to show that ligase E homologs are also upregulated following irradiation in *A. ricciae* and in the fungus *Mortierella verticillata*.

Perhaps the most significant result is determining whether AvLigE functions as a DNA ligase. Rather than knocking the AvLigE gene down or out, the authors did a ligation assay to show that protein extracts from irradiated *A. vaga* enhance DNA ligation activity, and more so when compared to extracts with AvLigE depleted. Lastly, they demonstrate that when AvLigE is expressed in human cells, this protein improves radiation tolerance and affected morphology of irradiated cells.

While many studies have speculated about the mechanisms of DNA repair in bdelloids and possible proteins involved in the process, this study is significant because it provides the first evidence of a protein found in bdelloid rotifers (and not in other rotifers, such as monogononts) that plays an important role in DNA repair following exposure to irradiation. While genetic manipulation of bdelloids is challenging, the authors use a creative approach with in vitro ligation assays and expression studies using human cells to show that AvLigE is a functional DNA ligase. Evolutionary analysis also shed light on the potential origin of AvLigE by horizontal gene transfer, shedding light on the importance of this process in driving evolutionary change.

The paper is well-written and the approach and methods for this study are carefully planned and well thought out. The authors make use of a variety of analytical approaches (proteomics, qPCR, phylogenetics, in vitro ligation assays, etc.) to address the central objective of understanding the function of ligase E in bdelloids. The paper should be considered for publication, and below are comments regarding specific details in the paper that should be addressed before that point is reached.

We are grateful to Reviewer #2 for appreciating the significance of our work and for pointing out the originality of the multidisciplinary approach that we have developed for this study.

Figure 1 legend: "Scheme A" should be "A."

Done.

Figure 1A: At t72h, "Gluthatione" should read "Glutathione"

Done.

Page 2, lines 25-28: The PFGE results are discussed first, followed by the proteomics (lines 28-33). However, in Figure 1 proteomics (1A) is shown before PFGE (1B). Consider reorganizing order of figures/text so that they better correspond.

Thank you for pointing out this layout issue. Since similar PFGE results have already been published elsewhere (e.g., Hespels et al., 2014; PMID: 25105197), they were removed from main Figure 1 and moved to the supplemental Figure S1A to focus on strictly novel data and to satisfy the correspondence between the text and the Figures. We now present the PFGE experiment as a way to track the level of IR-induced DNA damage at the doses used in the present study (Page 2, Lines 27-30).

Fig 1B: The PFGE shows striking evidence for DNA damage (at t0h) followed by DNA repair (at t4h, t24h and t72h). However, at t4h, t24h, and t72h, why does the repaired chromosomal DNA band not regain its original size (compared to the chromosomal band in the NI lane, which seems to be cutoff at the top of the gel, the dark bands in these lanes are smaller in size)? Does this imply that DNA repair is incomplete? Is there evidence that DNA repair is taking place accurately? For example, in this study do the animals recover from irradiation? Consider justifying this apparent discrepancy in the PFGE image when discussing Fig. 1B.

This is an interesting observation that confirms our previous PFGE analyses (e.g., Hespels et al., 2014 - PMID: 25105197, 2020 - PMID: 32849408 ; Terwagne et al., 2022 - PMID: 36449626) showing that DNA breaks were rapidly repaired after desiccation and IR exposure, but that a smear of DNA fragments persisted after prolonged (> 24h) recovery. Since these analyses were performed on whole animals, the results suggest that DNA DSB repair in the somatic nuclei of *A. vaga* is incomplete, leaving unassembled genomic fragments. In our recent paper (Terwagne et al., 2022, PMID: 36449626) aiming at studying the spatio-temporal dynamics of DNA repair in *A. vaga*, we found that all somatic nuclei became labeled with the thymine analog F-ara-Edu within 24 h after exposure to IR, consistent with fast repair of DNA lesions upon irradiation. Since adult bdelloid rotifers are eutelic, the somatic nuclei do not divide anymore and a rapid, yet partial repair of the fragmented DNA following IR maybe be sufficient to ensure cellular functions. In somatic nuclei, DSBs and other DNA lesions most likely involve base excision repair (BER) and nonhomologous end joining (NHEJ), repair pathways that predominates in non-dividing cells.

Our proteomic study here, together with recent transcriptomic analyses (Hecox-Lea and Mark Welch, 2018 - PMID: 30486781; Moris et al., in prep) reveal that NHEJ is likely not the only repair pathway that is activated upon IR treatment and that repairing other types of IR-induced DNA lesions may be crucial for stress recovery and survival. In particular, specific components of the base excision repair pathway (BER) such as polymerase β , PNKP, or the PARP proteins also appear to be up-regulated upon irradiation, albeit to a lesser extent than AvLigE-B (Page 2, Lines 35-36). Thus, as we discuss (Page 7, Lines 22-29), the timing and cellular localization of AvLigE-B expression support the conclusion that it may play a crucial role in rapid cellular recovery after IR exposure, but its specific contribution to one or the other DNA repair pathway remains to be determined.

Interestingly, Terwagne et al. (2022, PMID: 36449626) observed that DNA repair in the oocytes is delayed to a specific stage of oocyte maturation during which homologous chromosomes assemble into meiotic-like bivalents. The repair of the DNA DSBs in the germ line of *A. vaga* may therefore use the meiotic homologous recombination machinery to promote complete and accurate genome reassembly as was demonstrated by comparing the PFGE restriction profiles obtained for the parental line and the descendants of IR-treated mothers (Terwagne et al., 2022 - PMID: 36449626). Whether AvLigE also contribute to this germline-specific repair pathway or not requires further investigation.

We have now added the published results of Terwagne et al (2022, PMID: 36449626) at the end of the discussion (Page 7, Lines 22-29) and we added the following sentence “The smear remaining at late time points (t24h or t72h) suggests an incomplete DNA repair, leaving unassembled genomic fragments.” in the legend of Figure S1A to highlight the presence of the smear on PFGE following DNA repair.

Page 2, Line 37: Is there a citation for the observation that muscle contraction is visually affected in irradiated bdelloids? Or is that an observation from this paper? This is a very interesting detail, but it is unclear where this has been observed.

Same comment as reviewer #1 (see above). The new sentence in the manuscript (Page 2, Lines 40-43) provides more details on this intriguing 'behavioral' reaction to irradiation.

Page 2, Line 38: Earlier on page 2 (lines 32-33), the authors note that PNKP, DNA PolB and PARP proteins are upregulated following irradiation. Later (line 38), they state that all these proteins are upregulated at t24h and t72h. However, PARP3 is not included in the proteins that remain upregulated at t24h and t72h post-irradiation.

It is PARP2, not PARP3. In the new mass spectrometry analysis performed for the revision of the manuscript with the final protein database from Adineta vague published on NCBI (ASM2161353v1), PARP2 is present at all timepoints (t4h, t24h, and t72h).

Fig. 1D: It is surprising that DNA ligase-B is barely detectable in the t4h individual, especially since proteomic and western blot results suggest there should likely be more protein present. The authors note that immunofluorescence shows an increased nuclear signal at t24h which then decreases at t48h and t72h. However, the level of DNA ligase-B nuclear localization seems underwhelming, considering the total amount of DNA ligase-B signal at 24, 28 and 72h that does not localize to the nuclei. The authors should address the significance of the DNA ligase-B that is not localized to the nucleus. Also, DAPI (nuclei) staining is much brighter at t24h compared to other time points (e.g. non-irradiated)? Could some cellular process occurring during recovery from irradiation that would lead to this outcome?

We agree with Reviewer #2 that the data from Figure 1D (renamed Figure 1C in the revised version) was somewhat misleading. This is why we now provide a replicate of the immunostaining experiment in supplementary Figure S1D to testify the reproducibility of the results. In these experiments, the level of both AvLigE-B labeling and DAPI staining depends on the permeabilization of the rotifers and the position (and thus the accessibility) of the nuclei within the body of the animals after treatment (see details of the experimental procedure Page 9). This may explain some variability in the relative intensity of both signals. DAPI staining was rather strong at all timepoints, and we did not observe any significant increase of the signal intensity at t24h (Figure 1C, see also Figure S1D). At high resolution, AvLigE-B immunofluorescent signal overlapped with DAPI-stained somatic nuclei, which is globally reflected in the overlay images of Figure 1C and Figure S1D. We are thus confident of the primary nuclear localization of the DNA ligase AvLigE-B in *A. vaga*. However, we cannot exclude the possibility that some somatic nuclei express AvLigE-B at a higher level and/or for a longer period of time than others depending on the level of DNA repair that is required to satisfy specific cellular functions (see also answers to reviewer #3 below). Another striking example are the oocytes nuclei (now circled by dashed lines in Figure 1C and Figure S1D) in which no AvLigE-B expression is detected after IR treatment, as observed in the enlarged images of Fig S1D. As we now discuss Page 7, Lines 22-29, this correlates with our recent finding that DNA repair in germline nuclei

is postponed to a specific stage of oocyte maturation in order to fully reconstruct the genome by homologous recombination (see also above and answer to Reviewer #3's comments below).

Overall, the immunolocalization experiments shown in Figure 1C and new Figure S1D are consistent with the proteomic data demonstrating AvLigE-B expression after irradiation. As pointed by Reviewer #2 (and also Reviewer #3) there seems no strict correlation between the level of protein detection by western blot and the intensity of immunostaining signals observed by microscopy in these experiments. In the Western blot (Figure 1B), at t4h, DNA Ligase E copy B is not as strongly expressed as at t24h and t72h. For the immunofluorescence images, the signal of DNA ligase-B is indeed still very weak at t4h. At t24h, 48h and 72h the signal of DNA ligase-B is stronger (also in the Western blot) and co-localize with the somatic nuclei as revealed by DAPI staining. We believe that the observed differences may be due to technical reasons. In particular, the detection threshold and saturation level of both methods are likely to differ, which might explain the difference between the immunolocalization experiment and the western blot analysis (mostly at t4h). Therefore, quantitative interpretation of the immunostaining results is made more nuanced in the revised version of the manuscript (e.g., Page 3, Lines 13-18).

Figure 2: This is such an unusual grouping of taxa in the ligase E lineage. The authors do an excellent job addressing this with *Mortierella verticillata* and provide a novel analysis of radiation tolerance in this species and demonstrate that MvLigE is also upregulated following irradiation. However, is anything known about the DNA repair capacity of *Rotaria* sp.? Two *Rotaria* species have ligase E-B homologs but are noted as being desiccation-sensitive. A statement about why these two species would maintain ligase E would be helpful.

We agree with Reviewer #2 that the grouping of the taxa is unusual in the phylogeny.

The reason why the desiccation-sensitive *Rotaria* species retained the ligase E gene remains unclear (this is now specified in the text Page 3, Lines 38-39). Unfortunately, *Rotaria* samples do not grow under laboratory conditions, which precluded any functional analysis as we did for *Mortierella verticillata*.

Page 4 (lines 11-12): The authors state "To determine whether this horizontally acquired ligase may have a similar adaptive role as the one proposed for AvLigE..." Are the authors suggesting that MvLigE was also horizontally transferred in an independent event? If so, is there evidence from a similar Alienomics pipeline that was used for *A. vago*? Or this is based solely on the phylogenetic tree? If so, evidence from the tree alone needs to be better explained. The long branches (which could yield long branch attraction artifacts) and lack of bootstrap support for relationships within the ligase E lineage makes the claim of horizontal gene transfer difficult to understand. A similar issue is in the abstract (page 1, lines 16-17) where the authors state (...and its horizontal acquisition by bdelloid rotifers and other eukaryotes that also exhibit radiation tolerance...). They do not appear to provide evidence for horizontal transfer of ligase E in other eukaryotes (aside from the phylogenetic tree) and clarification needs to be addressed here.

We thank reviewer #2 for this important comment. Currently, the gold standard to detect HGT events is the phylogenetic approach. Taxonomic distribution of eukaryote species in our analyses (see Figure 2 and the additional Figure S2A) suggests potential HGT events of Ligase E in distinct eukaryotic clades.

We have tried to use Alienomics on *Mortierella verticillate* (Mv) available genomic data, but the results were inconclusive due to the low contiguity of the genome assembly (Alienomics is a tool that gains accuracy when used on highly contiguous genome assemblies). So, our interpretation is

solely based on the patchy distribution of eukaryotic taxa in the Ligase E phylogenetic tree (Figure 2 and S2A).

Based on the phylogenetic tree solely we cannot discriminate between a shared HGT event within bdelloid rotifers and *M.verticillata* or two independent events.

Regardless of the origin of the ligase E gene in *M. verticillata*, the over-representation of the single copy of the Ligase E gene in Mv upon exposure to ionizing radiation (>0.2 kGy) remains interesting and suggests a plausible correlation with radiation resistance, even if this indeed requires further investigations.

The lack of strong phylogenetic signal in the Ligase E phylogeny was also raised by reviewer #1 (see above). As detailed above, additional phylogenetic analyses were performed using sub-domains of the Ligase E gene (Figure S2B), still suggesting that the HGT hypothesis in bdelloid rotifers is the most likely and corroborating with the Alienomics results.

As we do not provide strong evidence for the horizontal acquisition of DNA Ligase E by other eukaryotes the word 'plausibly' was added in the abstract: "A phylogenetic analysis revealed its orthology to prokaryotic DNA ligase E, and its horizontal acquisition by bdelloid rotifers and plausibly other eukaryotes" (Page 1, Line 17).

Page 5 (lines 35-39): The authors mention other animals that have ligase E and note that they live in "marine coastal or shallow-water ecosystems that are stressful environments..." Clarification about what makes these environments stressful (in reference to DNA damage and the benefit of harboring a ligase E gene) would be helpful.

Same comment raised by Reviewer #1 (see above). We have added relevant references attesting the higher oxidative stress encountered in marine coastal habitats (Page 6, Lines 16-19), but we agree that its correlation with Ligase E acquisition remains speculative.

Fig. 3D. Unclear what this shows (the image with 6 panels and different levels of IR). Needs to be clearly explained in figure legend.

We provided more explanations for this panel in the legend of Figure 3.

Fig. 4D: Is there a significant difference between relative absorption when comparing the blue and green data points at each protein extract concentration? Some statistical analysis (as done for Fig. 5C) should be included to better demonstrate that the reduction in absorbance was significantly lower for the AvLigE-B depleted sample (compared to the IR sample) for corresponding extract concentrations.

Good point raised by Reviewer #2. A statistical analysis was performed to compare each timepoint to the control condition using Multiple t tests comparison on GraphPad Prism with a threshold for P value set to 0.05. The legends of the Figures and the Methods section were modified accordingly.

Fig. 5B. Same comment as for Fig. 4d (some sort of statistical analysis should be included).

Statistical analysis has been performed as for the answer to previous comment.

Fig. 5C: Could the increase in survival for cells expressing AvLigE be the result of overexpression of ligase? For example, if the authors included a cell line expressing ligase 1 from *A. vava* (or even a ligase from humans) and observed a similar result, this could suggest that overexpression of DNA ligase, in general, improves survival from irradiation. However, if they did that and only AvLigE

showed improved survival (as in Fig. 5E) then this would be more convincing evidence that AvLigE (and not simply ligase overexpression) specifically improves radiation tolerance. It is recommended that the authors include a trial with a ligase other than AvLigE to demonstrate the improved radiation tolerance is not an artifact of ligase overexpression.

Interesting comment from Reviewer #2. The difference in radiotolerance provided by AvLig E copy A in human cells compared to AvLig E copy B suggests a difference between Ligases. An overexpression of any ligase probably does not automatically improve the survival to irradiation, otherwise we should not have observed any difference between copy A and B.

Page 6, Line 7: “antioxydant’ should read “antioxidant”

Done. Page 6, Line 31.

Page 6, lines 37-50: The authors provide a compelling description for a mechanism of DNA repair in bdelloids. However, much of this speculation is based on unpublished findings from Terwagne et al. (via personal communication). For example, this section refers to somatic nuclei providing the strongest signal in the proteomic analysis (but this is not referred to in the manuscript) and a time window of DNA repair during oogenesis in which a meiosis-like configuration is present and different repair strategies for somatic and germline cells. These are intriguing findings but data to support these claims are not presented, not published elsewhere. Why were some of these data not included in this study to make for a more complete story of DNA repair? The authors should consider placing less emphasis on unpublished results at the end of their discussion section, or provide more data to support these statements.

Meanwhile, the manuscript of Terwagne et al. was published in Science Advances, end 2022 (PMID: 36449626), and we do refer to this manuscript in detail in the discussion section (Page 7, Lines 22-29).

Page 9, line 15: Three bdelloid genome assemblies are mentioned. However, the authors also note that ligase E homologs are absent in the monogonont *Brachionus plicatilis* genome. Is there an assembly for this?

We checked in our assembled genome of *Brachionus plicatilis* (not yet published) as well as in different public databases, but we did not find Ligase E in *B. plicatilis*.

Page 9, line 35: Was qPCR for bdelloids done in triplicate (as for *M. verticillata* mentioned below; line 47)?

Page 10, Lines 34 and 41 specifies that all qPCR experiments were performed in triplicates.

Reviewer #3 (Remarks to the Author):

The authors compare the proteome of the bdelloid rotifer *Adineta vaga* exposed to 1Gy of X irradiation with unexposed animals and find that a small number of proteins increase in abundance by > 2.5 fold. Among these is a protein encoded by a gene that is of non-Metazoan origin—one of the many “alien genes” that have been adopted by bdelloids. The authors characterize this gene as “ligase E–B,” one of a pair of genes differing in the number of PBZ domains (1 in “A” and 2 in “B”), with only the B protein being over-represented after irradiation. The authors support this result with qPCR of the two gene copies and of other ligases. They also show that a potential homolog in a fungus, with structural similarity to the A copy, is overexpressed after exposure to X-irradiation. The authors go on to use an antibody to the B protein to examine the localization of the protein in whole animals and to deplete cell lysates of the protein. They use a single-strand break assay to show that depleted lysates from irradiated animals have a decreased rate of repair compared to undepleted lysates. Finally, they transfect human tissue culture cells with the A and B copies and demonstrate that each increases the resilience of the cells to X-irradiation.

The application of proteomics to understand bdelloid radiation tolerance is a new and important advance, with significance beyond this particular ligase. The validation that this gene encodes a functioning ligase with a role in resilience to extreme DNA damage is the first proof of the hypothesis that this gene and other alien DNA repair genes acquired by bdelloids confer additional DNA repair capability. The demonstration that the ligase functions in human tissue culture cells is significant in its own right, and also opens up a new avenue for studying not only this ligase but the other proteins identified as playing a role in the remarkable radiation resilience of bdelloid rotifers.

We thank reviewer #3 for pointing out the broad interest of our work and for his/her constructive comments on the manuscript.

The gene was previously identified as a horizontally transferred ligase with potential to enhance bdelloid DNA repair in 2018 by Hecox-Lea and Mark-Welch (ref 27), who called the gene “LigK” with the same designation of A and B for the copies with one and two PBZ domains, respectively. They also presented a phylogenetic analysis demonstrating horizontal transfer into *A. vaga*, with a similar summary of domain structure in representative species, as well as RNA-Seq results showing that the transcript abundance of both copies increases in response to desiccation, with the B copy increasing about twice as much as the A copy. These points could be better leveraged in the manuscript.

We added this difference of expression observed by Hecox-Lea and Mark-Welch (2018) between B copy and A copy on Page 4, Line 25.

The statement in the abstract that “ligase E is by far the major contributor of DNA breaks ligation activity” is not supported by the evidence in the paper. Depletion of the B copy resulted in a reduction of ~35% in the rate of repair in a specific single strand break assay. While unquestionably significant, there is plenty of room for other major players even in this specific role, and of course there are many additional roles for ligases in DNA repair.

We agree with this comment of reviewer #3 and have refrained the sentence in the abstract by removing “by far the”.

Throughout the manuscript proteins are described as “up regulated” or “over expressed” or “differentially expressed” when proteomics or western blot shows an increase of protein levels. Given the numerous phenomena that could result in overabundance (change in transcript rate,

mRNA half-life, translation efficiency, protein half-life, etc) it would be more prudent to use a term like “over-represented” as the authors do in “Another protein, the alpha-protein kinase vwka, was also over-represented in the irradiated samples” or “induction” as used in the next paragraph.

We agree with Reviewer #3 and have changed the term "upregulated" proteins to "induced" or "over-represented" throughout the manuscript according to the context. When discussing the transcriptomic results or qPCR or heterologous expression of Av-LigE in human cells, the term "expression" was maintained.

The authors use a polyclonal antibody to the B protein. I do not see any demonstration of its specificity. Does it bind to the A protein? The A protein is ~51kD, a difference that can be resolved on the sort of gel shown in Fig4B. Can the gel image in Fig 4B be expanded to demonstrate whether the antibody binds to A? If it does bind to A I don't think this changes the overall significance of the results, the wording would just need to be tweaked to make it less specific.

Important comment from Reviewer #3. To address this question, we performed complementary experiments (presented in new Figure 1B and Figure S1C) using specific antibodies raised against DNA Ligases A and B, respectively:

1. After expression of AvLigE copy A or B in *E. coli*, the proteins were detected by Western-blot using the antibody specifically targeting copy A or copy B, or a poly-histidine tag fused to their C-terminal end. This new experiment (Figure S1C) demonstrates that each antibody more specifically recognizes its specific target with anti-DNA Ligase A recognizing copy A and anti-DNA Ligase B recognizing copy B.
2. To verify that the DNA Ligase copy A was not expressed after irradiation of *A. vago*, we performed a Western-blot analysis targeting DNA Ligase copy A, 24 h after irradiation (Figure 1B). This experiment demonstrates that DNA Ligase copy A is barely detected under non-irradiated condition (NI) and shows no induction after irradiation, confirming the proteomics results.

The authors find that human tissue culture cells transfected with copy A have greater repair activity than those with copy B, and in the Discussion speculate why the A protein would be more active than B in human cells. This seems to be based on the assumption that the A protein is less active than B in *A. vago*. But this may not be the case; the assays with *A. vago* only involved depletion of B. The A protein could be more active than B in *A. vago*. There is simply more of B produced in response to irradiation. Perhaps because it isn't as efficient as A.

We agree with Reviewer #3 that the role of AvLigE-A in *A. vago* remains unclear, notably because it is not or barely expressed under normal conditions and shows no induction upon IR exposure (as is now confirmed by the western blot analysis of new Figure 1B, see above). For this reason, we focused on the depletion of AvLigE-B to specifically highlight the impact of its IR-induced over-expression on radio-tolerance. However, as suggested by Reviewer #3 (and noted on Page 4, Lines 33-34), this does not rule out the possibility that AvLigE-A might contribute to the basal level of cellular DNA ligase activity as any other *A. vago* ligase.

However, because copy A is not expressed in *A. vago* after irradiation (see above) nor in the non-irradiated control (NI), it was difficult to test the effect of a depletion of copy A, nor to compare the relative activities of copies A and B in *A. vago*.

Fig 1A: "Protein with PARP domain" is listed only at 4 hours. In Fig S1 it is listed at 24 hours after 800Gy (Fig S1). Is it in the other timepoints below the arbitrary 2.5x threshold? Similarly PARP2 is present at 4 and 24 hours but not 72 hours, and not in Fig S1. Perhaps Table S1 could be mentioned in the legend or the results section and include more data? This would help solidify the contribution as the first proteomic investigation of bdelloid radiation exposure.

The raw data related to the proteomic analysis have now been posted on the PRIDE (Proteomics Identifications Database) repository and we added references to Table S1 in the legend of Figure 1 and in the Methods section.

The "Protein with PARP domain" (accession UJR36762.1) found at t4h is indeed only found at that timepoint. Even at a lower threshold, it was not found at the other timepoints. This may correspond to the specific expression of the protein at early timepoints, or be simply due to stochastic variations.

The PARP-2 protein (accession # UJR15000.1) was shown to be over-represented at 0.8 kGy vs 1.0 kGy of irradiation, showing the consistency of the results, and this protein was over-represented at the 3 timepoints (t4h, 24h and 72h) of 1kGy X-ray exposure (Figure 1).

Fig 1B: This is not an analysis, just a gel image. The legend should read something like "PFGE image showing DNA DSB repair following irradiation." Also, I'll point out here that this image and the general discussion of irradiation-produced DSBs is somewhat at odds with the data, which show that the ligase is capable of repairing single strand nicks. Of course there are presumably plenty of these nicks produced by 1kGy of X-rays, but the connection between this ligase and DSB repair is circumstantial.

The PFGE gel is moved to Figure S1 A (see also response to reviewer #2's comment above) and we have changed "PFGE analysis" to "PFGE image" in the legend of Figure S1.

We totally agree with the suggestion made by Reviewer #3 that DNA AvligE is not specifically or exclusively involved in the repair of DSBs. Formation and resolution of DSBs is used here as a readout for IR-induced DNA damages and the dynamics of DNA repair processes in *A. vaga*. As we now discuss in detail at the end of the manuscript (Page 7, Lines 22-29), our data support the conclusion that AvLigE could be more broadly involved in the rapid repair of somatic DNA lesions arising from genotoxic stresses such as high-doses of IR exposure as revealed by Fara-EdU labeling in our recently published paper (Terwagne et al., 2022 - PMID: 36449626). This may reflect the implication of AvLigE in different repair mechanisms involving a DNA ligation step, and its overall contribution to cell survival. Supporting this view, we show here that IR-induced expression of AvLigE-B coincides with the induction of other key actors of the base excision repair (BER) pathway that is specifically required to remove damaged base pairs (see also the answer to reviewer #2's comment above).

Fig 1D: I don't know what to make of this figure. It does show the presence of the protein <in situ over a time course consistent with Figs 1A and C, and presumably shows that the B protein is confined to nuclei. But in the overlay images I'm not seeing anything like a 1:1 correspondence between DAPI and immunofluorescence. There seem to be red spots without corresponding green spots and vice versa. Green without red could be simply brighter green signal, but it looks like there are entire organs without B protein. Could expression of copy B be confined to certain cells? That would certainly be interesting!

Also, only one image of each time point is shown so it is impossible to evaluate the assertion that signal "appears to decrease at t48h and t72h." I would point out that this interpretation is at odds with the increase in the abundance of copy B shown in Figs 1A and C.

The purpose of this experiment was to define the subcellular localization of DNA Ligase copy B at different timepoints. This is the reason why we focused on single individuals to have a better resolution. Because of the variable level of permeabilization of the rotifers due to thickness of the cuticle, it is difficult to perform accurate quantitative analysis on these microscopy images. However, we understand the point of the reviewer, this comment was also raised by reviewer #1 (see above), and we provided for each timepoint another image in Figure S1D showing similar results as Figure 1C. We also provide enlargements of some images in Figure S1D to further clarify the different observations between somatic and germline nuclei.

For the immunofluorescence images in Figure 1C (1D became 1C in the new figure), at t24h and 48h the signal of DNA ligase-B is stronger than at t4h (corroborating with the results of the Western blot analysis) and localizing at the somatic nuclei when compared to the DAPI staining. In the overlay images at t24h and t48h both the DNA ligase-B and DAPI signals overlap, but the relative intensity of the signals (red and green) differs and therefore, one color sometimes appears less visible than the other in the overlay image. When checking the staining separately we see that the red and green signal stains the same somatic nuclei except in the regions of the ovaries (that we now have encircled by dashed lines in Figure 1C and Figure S1D). This agrees with the observation made by Terwagne et al. (PMID: 36449626), that the oocytes in the ovaries are not stained with F-ara-Edu (a marker of DNA-repair associated DNA synthesis) during or after exposure to ionizing radiation. Instead, DNA repair in the oocytes is delayed to a later stage during which homologous chromosomes were found to adopt a meiotic-like juxtaposed configuration. In the somatic nuclei however, Terwagne *et al.* (2022 - PMID: 36449626) observed that the DNA repair occurred rapidly with all nuclei being stained with F-ara-Edu immediately after exposure to IR. The repair of the DNA DSBs in the germ line of *A. vaga* is thought to occur by homologous recombination to ensure complete genome reassembly, while in the somatic nuclei, DSBs are most likely to be sealed by nonhomologous end joining (NHEJ) that generally predominates in non-dividing cells.

In the result section we now specify this absence of labeling of the ovaries by DNA ligase E copy B. We added the following sentences (Page 3, Lines 15-17): “However, virtually no DNA ligase B labeling was observed in the middle part of the adult body where the ovaries and germline are localized (circled by dashed lines in Figure 1C and Figure S1D). The IR-induced expression pattern of this DNA ligase B and its nuclear localization in *A. vaga*, strongly suggests that it might be a key actor involved in bdelloid rotifer DNA repair in somatic nuclei”. In the discussion (Page 7, Lines 22-29) we discuss this in light of the results of Terwagne et al. (2022 - PMID: 36449626) showing that DNA repair in the somatic line and the germline of *A. vaga* is uncoupled (or takes place with different timing and likely different mechanisms). It is indeed interesting to observe here that this Ligase E copy B seems to be confined to somatic cells in *A. vaga* and that our results corroborate with those published by Terwagne et al. (2022).

Fig 2: I am not convinced that the “Ligase E” clade is meaningful. This is an unrooted tree and there is no clade containing the eponymous bacterial Ligase E and the other sequences labeled as “Ligase E”; the only support is that they are all not in the other ligase clades. There is as much sequence diversity in the proposed clade as there is between all other ligase clades combined, and unlike other ligase families there are drastically different domain architectures within the clade, implying very different properties and activities (the canonical Ligase E gene from bacteria contains only the two domains colored blue and green in the figure, and a signal peptide for export, and is associated primarily with repair of nicks). So in addition to not being a true clade it is unlikely to be a useful distinction that will help us designate or understand ligases. I agree that the original name of “Ligase K” is also not ideal; it was chosen because all the genes designated LigK in ref 27 had top hits to conserved domains Adenylation_kDNA_ligase_like (cd07896) and OBF_kDNA_ligase_like (cd08041) in the NCBI CDD database available at the time, but this isn’t true of newer database releases.

That the ligase genes are alien was already demonstrated in ref 27 (albeit with fewer sequences) so the figure is not necessary for that purpose. The figure does include three additional bdelloid species, which helps identify the horizontal transfer as a single event early in bdelloid evolution, an important point the authors note. It also includes the protozoan *P. fungivorum*, which is the closest known sequence and the only non-bdelloid example of the PBZ domain architecture seen in the B copy (though the additional peroxidase domain is certainly unusual). When I look at the gene tree I see phyletic incongruence, patchy distribution, and low resolution, implying that the core blue/green domains have been passed around and adapted for different purposes by different lineages, which brings into question the whole concept of a meaningfully "homologous" gene family.

We are not sure to fully understand the point raised here by Reviewer #3, but we will try to clarify some aspects.

The sequences that we designated as "Ligase E" form a monophyletic group (73% bootstrap, Figure 2), and therefore the Ligase E clade appears meaningful from an evolutionary perspective. The tree is rooted here on other ligases. The observed diversity in each ligase clade depends on how many sequences are included. For the "other Ligases" (i.e. all Ligases except Ligase E) this was only a few sequences (see Figure S2 that we added with all sequences highlighted). Domain architecture within the Ligase E clade is stable for both the canonical ATP domain and OB domain in all species (with additional architecture variability being detected among species). In our case, the top blast hit for AvLigE is Ligase E. We argued above that "Ligase K" might actually correspond to a fast-evolving "Ligase E" within the kinetoplastids (see answer to reviewer #1's comment above, and the additional phylogenetic trees provided in Figure S2).

As we now clearly point out in the text (Page 2, Line 49 and Page 3, Lines 43-44), HGT origin of AvLigE was indeed previously suggested (Hecox-Lea & Mark Welch, 2018 - PMID: 30486781), but Figure 2 is important to visualize that the HGT acquisition of Ligase E is ancient and likely occurred as a single event in the ancestor of all bdelloid rotifers. Ligase E evolution is indeed complex and difficult to accurately infer: taxonomic distribution is patchy and phylogenetic signal is often weak. No strong incongruence is however observed between the different analyses that we performed during this revision on the different domains of the Ligase E gene (see response to reviewer #1 above and Figure S2). In all these analyses, Ligase E sequences were monophyletic, and the internal branch leading to the clade of ligase E + ligase K is long, indicating that the ligase E is a meaningful, divergent lineage of ligases. We agree however with Reviewer #3 that the function of these ligases E might differ from one lineage to another. A "homologous gene family" describes an evolutionary origin (not necessarily a function).

Fig 3A: In the context of this panel the authors may want to mention that desiccation also increases transcript abundance of copies A and B, with B increasing twice as much as A, but that the magnitude of the change in Ligase B is *much* greater after irradiation. This connects the irradiation work done here with the more biologically relevant process of desiccation.

We thank the reviewer for this interesting comment and added the following sentences in the result section (Page 4, Lines 24-26): « In desiccated *A. vaga* individuals an increase of transcript abundance of both copies A and B was detected, with B increasing twice as much as A (Hecox-Lea *et al.*, 2018 - PMID: 30486781), while after irradiation we barely detected copy A of Ligase E in both bdelloid species tested and only copy B was strongly expressed ».

Methods:

"were lyzed by 1,000 Dounce with tight pestle" missing a word?

We corrected this by: rotifers were lyzed by 1,000 Dounce **pulses** with tight pestle (Page 8, Line 3).

“proteic” isn’t a common word in English. Maybe “peptide”?

We changed it by protein throughout the manuscript

Reviewer #4 (Remarks to the Author):

In this study Nicolas and colleagues characterized a prokaryotic DNA ligase involved in DNA damage tolerance. Mass spectrometry proteomic analysis of a rotifer organism irradiated for different time points led to the identification of a DNA ligase as the most, according to the authors, upregulated protein post-irradiation. Other proteins with potential roles in regulating DDR were also identified in the screen. The analysis was then focused on characterising DNA ligase expression in related species and in testing the hypothesis that this enzyme confers tolerance to DNA damage by heterologous expression in human cells.

Major issues

1. The analysis and dissemination of proteomics data does not conform to standards in the field. The mass spectrometry raw data has not been submitted to a repository (e.g. PRIDE) and processed data for all identifications (identities, sequences and LF quantity values for all identified peptides) should be provided in supplementary datasets. Without these data, it is not possible to assess the quality of data that is reported in the article.

The raw and processed proteomics data have been deposited on the PRIDE repository as proposed by the reviewer.

Reviewer account details:

Username: reviewer_pxd043051@ebi.ac.uk

Password: LR4iOLGO

2. There are no information on the database that was used for protein identification from mass spectrometry data. How many sequences does it contain and how was were the sequences translated?

More information is now provided on the Methods section (Page 8, Lines 42-43): "The peak lists were searched against the protein database from *Adineta vaga* (NCBI, assembly ASM2161353v1) containing 33,103 entries."

3. The proteomic experiment seems to have been carried out once without technical or biological replicates.

As given in the Methods section (Page 7, Line 43; Page 8, Line 37), each irradiation and subsequent MS analysis was performed in duplicates (two biological replicates) and two injections per sample were done (two technical replicates).

4. There is no justification for the threshold for considering differentially regulated proteins to be set to 2.5 fold.

We did choose a threshold of 2.5-fold for our analysis because below this threshold the response to irradiation gave many unique accessions at different timepoints. Therefore, to highlight the most upregulated proteins after irradiation, we decided to use a threshold of at least 2.5-fold because this was the threshold at which the same proteins were consistently over-represented at the different timepoints. This explanation is clarified in the Methods section (Page 8, Line 51; Page 9, Line 1).

5. There are no details of how the normalization of the LFQ data was carried out.

More details regarding the normalization of LFQ data are provided (Page 8, Lines 44-50)

6. The Supplemental tables are not informative. For example, data in Table S1 is not informative because the acronyms in the headings are not defined in a table heading or caption.

We would like to keep this Table S1 in the manuscript because it provides more information than the list provided in Figure 1. However, we simplified it in comparison to the one provided during the first submission of the article because the raw data are now also submitted on the PRIDE database. We also provided more details related to the names of the headings as footnote of the Table S1.

REVIEWER COMMENTS

Reviewer #1 (Remarks to the Author):

Thanks for the authors for their thorough revisions. There are just a couple of minor remaining points that I think could be clarified.

First, for the response: We added the following information on Page 3, Lines 30-31: "Phylogenetic analyses revealed that members of this family are mainly found in bacteria (> 50% of species collapsed at the branch "Bacteria" in Figure 2 and >69% are bacteria in Figure S2)".

Mainly still doesn't sound quite right word for 78 bacteria versus 77 everything else. Could say "Phylogenetic analyses revealed that over half of the members of this family are found in bacteria (> 50% in Figure 2 and >69% in Figure S2)"

Second, in a couple places the text seems to imply transfer to bdelloids from bacteria, whereas the results presented strongly suggest transfer from some kind of protozoa/fungi/chromista instead.

1) Page 3 line 25. "the BLASTp analysis showed that the most closely related sequences found in public databases do not belong to other metazoans but well to members of other eukaryote groups (i.e., Amoebozoa, fungi, Ciliophora and Chromista) and to prokaryotes (Table S2)".

The last part of this sentence seems inaccurate. Table S2 doesn't report the taxonomic identity of the closest match. Both figure 2 and S2 show that the bdelloids are closer to the other animals than they are to any bacteria. Suggest delete "and to prokaryotes" in this sentence.

2) Line 46 page 5: "Phylogenetic analyses also revealed that the closest orthologous ligase E sequences of bdelloid rotifers are found in the groups of Protozoa, Fungi, and Chromista (Figure 2), suggesting that similar horizontal transfers of ligase E might have occurred independently in different phyla to adapt to extreme genotoxic stresses"

The more parsimonious explanation is that bdelloids received their copy by HGT from another eukaryote, being some kind of protozoa/fungi/chromista. The phylogenetic tree suggests this, the presence/absence of particular eukaryotic domains shared with those taxa also suggests it.

Reviewer #2 (Remarks to the Author):

Much appreciation to the authors for the thoughtful and detailed responses to questions and comments in the first evaluation of the paper. The majority of issues and requests for clarifications were addressed. In particular, the discussion about immunostaining was very informative (the inclusion of dashed lines to delimit the oocytes from somatic cells is an intriguing observation and correlates with the Terwagne et al paper) and the authors provided informative explanations and helpful edits to address questions about the phylogenetic analysis, inclusion of statistics for Figs. 4D and 5B, and several other points. Below are a few outstanding questions and comments:

Fig 1B: The description of the current state of understanding of DNA repair in bdelloids is appreciated and explains why smearing is still present following DNA repair. The original main issue with Fig. 1B was the faint appearance of the bands in the NI lane and why the corresponding band of reassembled DNA in subsequent lanes is so much darker. Does this simply mean more rotifers were used in the irradiated samples (darker chromosomal DNA bands) compared to the NI lane (fainter chromosomal DNA band), and this difference in intensity on the gel is simply a matter of DNA quantity? Or is there chromosomal DNA at the top of the gel in lane NI that was cut off in the figure (there is a dark band at the very top; is this the well in the gel?)

Page 2, line 36: If PARP2 is present at all timepoints, why not simply say "PARP2" here rather than "PARP-domain containing proteins"? This point is made because the authors note that PARP-

containing proteins are found at t4h, but it is only PARP2 that is found at all time points. On line 43 the authors state "All these proteins remained over-represented at t24h and t72h..." which implies all of the PARP-domaining containing proteins at t4h, but this is not accurate since only PARP2 was found at all time points.

Reviewer #3 (Remarks to the Author):

The authors have addressed all my concerns in this improved manuscript. I have only a few comments that do not, in my opinion, require another round of review:

1. I don't see the very helpful new Figure S1C referenced in the text.
2. I agree with reviewer 1 that the relationship to the bacterial lig E family is overstated. The statement on p2 line 19 that the ligase "has strong homologies with the ligase E family of prokaryotic DNA ligases" isn't supported by Fig 2 or Fig S2, as the closest relatives are eukaryotic and bootstrap support to the bacterial LigE clade is non-existent. The new analysis of Fig S2 only reinforces the evidence that the gene is of eukaryotic origin. The fact that "members of this family are mainly found in bacteria (> 50% of species collapsed at the branch "Bacteria" in Figure 2 and $\geq 69\%$ are bacteria in Figure S2A)" is irrelevant: this statement is true of nearly any gene shared across domains, given that so many more bacterial than protistan genomes have been sequenced and that there are so many more bacterial genomes in nature. If the authors want to continue to assert this that is up to them.
3. Can table S3 be made landscape or reorganized such that the group numbers can be assigned to the rest of the table? Perhaps the whole thing will be an excel file (or better yet tab delimited or csv) in final form?

Reviewer #4 (Remarks to the Author):

I thank the authors for their efforts in submitted to the PRIDE repository and for providing more information on how the proteomics experiments were carried out. However, the data in PRIDE is not complete. For example, the identification files contain just one entry (one row) and supplementary datasets are not included with this resubmission. The authors should check that their datasets are complete so that the paper can be reviewed and interested readers can assess the study after publication.

Reviewer #1 (Remarks to the Author):

Thanks for the authors for their thorough revisions. There are just a couple of minor remaining points that I think could be clarified.

First, for the response: We added the following information on Page 3, Lines 30-31: "Phylogenetic analyses revealed that members of this family are mainly found in bacteria (> 50% of species collapsed at the branch "Bacteria" in Figure 2 and >69% are bacteria in Figure S2)".

Mainly still doesn't sound quite right word for 78 bacteria versus 77 everything else. Could say "Phylogenetic analyses revealed that over half of the members of this family are found in bacteria (> 50% in Figure 2 and >69% in Figure S2)"

> We did modify the text accordingly. Page 3, Line 29.

Second, in a couple places the text seems to imply transfer to bdelloids from bacteria, whereas the results presented strongly suggest transfer from some kind of protozoa/fungi/chromista instead.

1) Page 3 line 25. "the BLASTp analysis showed that the most closely related sequences found in public databases do not belong to other metazoans but well to members of other eukaryote groups (i.e., Amoebozoa, fungi, Ciliophora and Chromista) and to prokaryotes (Table S2)".

The last part of this sentence seems inaccurate. Table S2 doesn't report the taxonomic identity of the closest match. Both figure 2 and S2 show that the bdelloids are closer to the other animals than they are to any bacteria. Suggest delete "and to prokaryotes" in this sentence.

> We agree with Reviewer #1 and deleted "and to prokaryotes" from the text as suggested. Indeed, the HGT may rather come from a fungus or protozoan. Page 3, Line 25-26.

Allusion to the prokaryotic origin of Ligase E family members has also been removed from the title of the paper to avoid any confusion (Page 1). Revised version of the title reads "Horizontal acquisition of a DNA ligase improves DNA damage tolerance in eukaryotes" (see also comment from Reviewer #3 below)

2) Line 46 page 5: "Phylogenetic analyses also revealed that the closest orthologous ligase E

sequences of bdelloid rotifers are found in the groups of Protozoa, Fungi, and Chromista (Figure 2), suggesting that similar horizontal transfers of ligase E might have occurred independently in different phyla to adapt to extreme genotoxic stresses”

The more parsimonious explanation is that bdelloids received their copy by HGT from another eukaryote, being some kind of protozoa/fungi/chromista. The phylogenetic tree suggests this, the presence/absence of particular eukaryotic domains shared with those taxa also suggests it.

> We agree with this reviewer and changed the text accordingly, Page 5, lines 46-48.

Reviewer #2 (Remarks to the Author):

Much appreciation to the authors for the thoughtful and detailed responses to questions and comments in the first evaluation of the paper. The majority of issues and requests for clarifications were addressed. In particular, the discussion about immunostaining was very informative (the inclusion of dashed lines to delimit the oocytes from somatic cells is an intriguing observation and correlates with the Terwagne et al paper) and the authors provided informative explanations and helpful edits to address questions about the phylogenetic analysis, inclusion of statistics for Figs. 4D and 5B, and several other points. Below are a few outstanding questions and comments:

Fig 1B: The description of the current state of understanding of DNA repair in bdelloids is appreciated and explains why smearing is still present following DNA repair. The original main issue with Fig. 1B was the faint appearance of the bands in the NI lane and why the corresponding band of reassembled DNA in subsequent lanes is so much darker. Does this simply mean more rotifers were used in the irradiated samples (darker chromosomal DNA bands) compared to the NI lane (fainter chromosomal DNA band), and this different in intensity on the gel is simply a matter of DNA quantity? Or is there chromosomal DNA at the top of the gel in lane NI that was cut off in the figure (there is a dark band at the very top; is this the well in the gel?)

> This is a recurrent and quite reproducible observation from irradiation and desiccation experiments (e.g., Hespels et al., J Evol Biol 2014; Front Microbiol 2020; BMC Biol 2023). In these experiments, the same number of rotifers (~ 1000) was embedded within the agarose plugs for both the treatment and control conditions before to start the PFGE. We believe that the DNA signal appears much stronger with the irradiated samples than under the NI condition because the IR samples contain fragmented DNA that can more easily diffuse out of the cells and migrate into the gel than intact chromosomes of NI individuals which tend to get stuck into the wells (and eventually diffuse away during the migration). Supporting this explanation, retention of DNA into the wells is not observed when embedded NI samples are treated with restriction enzymes prior to PFGE, giving a discrete pattern of well separated genomic fragments (e.g., see Terwagne et al., Sci Adv 2022).

To clarify this point, we added the following sentence to the legend of Figure S1: “The weak signal observed for the NI samples likely results from the fact that intact *A. vaga* chromosomes are too big to enter into the gel”.

Page 2, line 36: If PARP2 is present at all timepoints, why not simply say “PARP2” here rather than “PARP-domain containing proteins”? This point is made because the authors note that PARP-containing proteins are found at t4h, but it is only PARP2 that is found at all

time points. On line 43 the authors state “All these proteins remained over-represented at t24h and t72h...” which implies all of the PARP-domain containing proteins at t4h, but this is not accurate since only PARP2 was found at all time points.

> We agree with this reviewer, only PARP2 was found to be overexpressed at all timepoints. We therefore changed on Page 2 Line 36 “PARP-domain containing proteins” by PARP2.

Reviewer #3 (Remarks to the Author):

The authors have addressed all my concerns in this improved manuscript. I have only a few comments that do not, in my opinion, require another round of review:

1. I don't see the very helpful new Figure S1C referenced in the text.

> Typo on Page 3, Line 7. Figure S1B needs to be changed to Figure S1C.

2. I agree with reviewer 1 that the relationship to the bacterial lig E family is overstated. The statement on p2 line 19 that the ligase “has strong homologies with the ligase E family of prokaryotic DNA ligases” isn't supported by Fig 2 or Fig S2, as the closest relatives are eukaryotic and bootstrap support to the bacterial LigE clade is non-existent. The new analysis of Fig S2 only reinforces the evidence that the gene is of eukaryotic origin. The fact that “members of this family are mainly found in bacteria (> 50% of species collapsed at the branch “Bacteria” in Figure 2 and \geq 69% are bacteria in Figure S2A)” is irrelevant: this statement is true of nearly any gene shared across domains, given that so many more bacterial than protistan genomes have been sequenced and that there are so many more bacterial genomes in nature. If the authors want to continue to assert this that is up to them.

> There are two important aspects here to clarify.

1) On Page 2 Line 19 we mention “We provided evidence that this enzyme, renamed AvLigE, has strong homologies with the ligase E family of prokaryotic DNA ligases, and that it was horizontally acquired ...”. This seems correct since AvLigE shares strong homologies with the ligase E family of prokaryotes.

2) However, as stated in our response to Reviewer #1 above, we agree that the closest relatives to AvLigE are non-metazoan eukaryotes, and that it is more likely that the ancestor of bdelloid rotifers acquired their LigE from an eukaryote rather than a prokaryote. We therefore removed “prokaryotes” from the title and changed the sentence on Page 5; Lines 46-48 by “This may suggest a horizontal transfer of Ligase E from an eukaryotic non-metazoan group to an ancestor of bdelloid rotifers, ...”

3. Can table S3 be made landscape or reorganized such that the group numbers can be assigned to the rest of the table? Perhaps the whole thing will be an excel file (or better yet tab delimited or csv) in final form?

> Tables have been submitted in Excel format. The reviewers maybe only received a pdf format of these files? This will be verified with the editor after final approval of the manuscript.

Reviewer #4 (Remarks to the Author):

I thank the authors for their efforts in submitted to the PRIDE repository and for providing more information on how the proteomics experiments were carried out. However, the data in PRIDE is not complete. For example, the identification files contain just one entry (one row) and supplementary datasets are not included with this resubmission. The authors should check that their datasets are complete so that the paper can be reviewed and interested readers can assess the study after publication.

We believe that the dataset deposited on PRIDE is now complete. Some identification files indeed only contain one entry (e.g. protein-peptides-1000Gy-LFQ-0h.csv), which is in accordance with our results since only one protein was differentially over-represented in irradiated conditions at this timepoint. To the contrary, files corresponding to later timepoints (e.g., proteins-1000Gy-LFQ-72h.csv) contain more rows (64 in that case) as more proteins were differentially expressed at this timepoint when comparing non-irradiated and irradiated conditions. We do hope this has clarified this point.

The supplementary dataset is now included in this new resubmission as requested by the reviewer.